# *JAK2V617F* mutation drives vascular resident macrophages toward a pathogenic phenotype and promotes dissecting aortic aneurysm

Rida Al-Rifai [1,13], Marie Vandestienne[1,13], Jean-Rémi Lavillegrand[1], Tristan Mirault[1,2], Julie Cornebise[1], Johanne Poisson[1,3,4], Ludivine Laurans[1], Bruno Esposito[1], Chloé James[5], Olivier Mansier[5], Pierre Hirsch[6], Fabrizia Favale[6], Rayan Braik[1], Camille Knosp[1], Jose Vilar [1], Giuseppe Rizzo[7], Alma Zernecke[7], Antoine-Emmanuel Saliba [8], Alain Tedgui [1], Maxime Lacroix[9], Lionel Arrive[9], Ziad Mallat[1], Soraya Taleb [1], Marc Diedisheim [10], Clément Cochain[7], Pierre-Emmanuel Rautou[1,4,11] & Hafid Ait-Oufella [1,12] ✉

JAK2V617F mutation is associated with an increased risk for athero-thrombotic cardiovascular disease, but its role in aortic disease development and complications remains unknown. In a cohort of patients with myeloproliferative neoplasm, *JAK2V617F* mutation was identified as an independent risk factor for dilation of both the ascending and descending thoracic aorta. Using single-cell RNA-seq, complementary genetically-modified mouse models, as well as pharmacological approaches, we found that *JAK2V617F* mutation was associated with a pathogenic pro-inflammatory phenotype of perivascular tissue-resident macrophages, which promoted deleterious aortic wall remodeling at early stages, and dissecting aneurysm through the recruitment of circulating monocytes at later stages. Finally, genetic manipulation of tissue-resident macrophages, or treatment with a Jak2 inhibitor, ruxolitinib, mitigated aortic wall inflammation and reduced aortic dilation and rupture. Overall, *JAK2V617F* mutation drives vascular resident macrophages toward a pathogenic phenotype and promotes dissecting aortic aneurysm.

Aortic aneurysm (AA) is a frequent vascular disease affecting approximately 5% of elderly men in Western countries[1] whose incidence is still increasing despite better control of risk factors like smoking and blood pressure. Surgery and endovascular procedure are the only treatments for individuals with aortic aneurysm to limit the risk of aortic rupture and dissection, but the interventions are costly and associated with high morbidity and mortality[2]. Currently, no drug treatments have been approved for use in aortic aneurysm, which underlines the need for a better understanding of mechanisms underlying the disease in order to implement novel management procedures and therapeutic strategies.

The mechanisms of aortic aneurysm formation are complex, including elastin degradation, collagen remodeling, smooth muscle cell apoptosis and local activation of immune cells such as macrophages[3]. Several experimental studies have suggested that during the first stages of AA formation, circulating monocytes interact with activated endothelial cells (ECs) and are recruited into the aortic wall, Ly6C[high] monocytes dominating the acute phase of injury while non-classical Ly6C[low] monocytes being prevalent thereafter[4]. Recruitment into the vascular wall depends, at least in part, on GM-CSF[5], chemokines/chemokine receptors[6], and adhesion molecules.

---

Monocytes locally differentiate into macrophages and, after activation, promote aneurysm formation and complications (dissection and/or rupture) through the release of inflammatory cytokines (including TNF-α, IL-1β) and matrix metalloproteinases (MMPs). Another subset of macrophages, tissue-resident macrophages, which originate from the yolk sac and fetal liver during development[7,8], has been recently identified as essential for the maintenance of arterial wall homeostasis through the regulation of smooth muscle cell activity and collagen production[9]. Tissue-resident macrophages express the lymphatic vessel endothelial hyaluronan receptor-1 (LYVE-1), but their role in aortic aneurysm formation and progression remains unknown.

*JAK2V617F* is a gain-of-function mutation responsible for enhanced hypersensitivity to growth factors and cytokines[10]. *JAK2V617F* mutation promotes the proliferation, the survival and the maturation of hematopoietic progenitors leading to myeloproliferative neoplasms such as polycythemia vera, essential thrombocythemia, and primary myelofibrosis[10,11]. The presence of *JAK2V617F* mutation is associated with an increased risk of cardiovascular diseases[12]. Experimental studies have shown that *JAK2V617F* mutation in bone marrow cells, mimicking human myeloproliferative neoplasms, accelerates atherosclerosis in hypercholesterolemic mice due to impaired red blood cell efferocytosis and inflammasome activation[13], and aggravates post-ischemic cardiac remodeling[14]. However, the role of *JAK2V617F* mutation in aortic aneurysm development and complications has never been investigated.

Here, we showed that *JAK2V617F* mutation is associated with aortic dilation in patients. Moreover, using several complementary genetic and pharmacological approaches in murine models, we found that *JAK2V617F* mutation drives adventitial tissue-resident macrophages toward a pathogenic inflammatory phenotype causing aortic aneurysm and dissection.

## Results

### Global JAK2V617F mutation promotes aortopathy in humans, and aortic aneurysm and dissection in mice

Several epidemiological studies have shown that *JAK2V617F* gain-of-function mutation is associated with increased risk for athero-thrombotic cardiovascular disease, but to our knowledge nothing has been reported regarding the possible association between this mutation and aortic disease. To address this issue, we first screened *JAK2V617F* + patients recruited at Saint-Antoine Hospital ($N = 382$, Paris, France), and identified those who had a recent Body CT-scan ($N = 157/382$). Mean diameter was measured in different segments of the aorta (ascending, descending thoracic and abdominal) and compared to age- and gender-matched controls by a blinded independent reader (Supplementary Fig. 1). General characteristics of the cohort are detailed in Supplementary Table 1. Interestingly, after multiple adjustment on age, gender, hypertension, diabetes and smoking, we found that the *JAK2V617F* mutation was associated with significant dilation of both ascending and descending thoracic aorta, but not abdominal aorta (Fig. 1A, B).

In order to understand the cellular and molecular mechanisms responsible for aortopathy in *JAK2V617F*+ patients, we used genetically-modified mouse models. First, we studied Jak2V617F^WT/Flex VE-Cadherin^-Cre+/- mice, herein referred to as *Jak2^V617F HC-EC* mice, in which Cre-mediated recombination occurs in ECs and hematopoietic cells (HCs) due to the expression of VE-cadherin during early embryonic life in a precursor of both ECs and HCs[15,16] (Supplementary Fig. 2A). *Jak2^V617F HC-EC* mice developed myeloproliferative neoplasm, as attested by palmar erythema (Fig. 1C), splenomegaly ($87 \pm 27$ vs $560 \pm 81$ mg; $P < 0.0001$) (Fig. 1D), and higher red blood cell, platelet and leukocyte counts than littermate controls (Fig. 1E & Supplementary Fig. 2B). At 7 weeks, we performed histological analysis and we found aortic dilation in *Jak2^V617F HC-EC* mice affecting descending thoracic ($0.71 \pm 0.07$ vs $0.80 \pm 0.08$ mm, $P < 0.05$) and

abdominal ($0.74 \pm 0.03$ vs $0.85 \pm 0.08$ mm, $P < 0.01$) aortic regions (Fig. 1F). Notably, both systolic and diastolic blood pressure were significantly lower in *Jak2^V617F HC-EC* mice than in littermate controls, which rules out the possibility that aortic dilation resulted from hypertension (Supplementary Fig. 2C).

At 7 weeks of age, hematopoietic and endothelial *JAK2V617F* gain-of-function mutation was associated with enhanced elastin degradation, as assessed by the significantly lower number of elastin layers in the abdominal aorta of *Jak2^V617F HC-EC* mice compared with the aorta of control *Jak2^WT HC-EC* mice ($3.90 \pm 0.12$ vs $3.70 \pm 0.17$ layers, $P < 0.05$) (Fig. 1G), and reduced content in collagen type 1a (Fig. 1H & Supplementary Fig. 2D), but no difference in α-SMA + smooth muscle cell content (Supplementary Fig. 2E). The decrease in extracellular matrix protein content in *Jak2^V617F HC-EC* mice was accompanied with an increase in global MMP activity in the aortic wall, quantified using ex vivo fluorescence molecular tomography imaging (Fig. 1I) and qPCR (Supplementary Fig. 2F). Interestingly, we observed specifically in *Jak2^V617F HC-EC* mice spontaneous early death (Fig. 1J & Supplementary Fig. 2G), reaching 70% at 20 weeks of age; which was due to aortic dissecting aneurysm (Fig. 1K) and thoracic and/or abdominal hemorrhage. Polyglobulia correction using phenylhydrazine[17] in *Jak2^V617F HC-EC* mice did not impact either on aorta dilation or rupture incidence (Supplementary Fig. 3).

### JAK2V617F mutation specifically expressed in vascular cells did not induce dissecting aortopathy

To assess the implication of *JAK2V617F* mutation in vascular cells, we used 2 complementary approaches. First, we generated chimeric mice expressing *JAK2V617F* mutation only in non-hematopoietic cells by transplanting C57BL/6 bone marrow (BM) cells into lethally irradiated *Jak2^V617F HC-EC* mice and C57BL/6 controls (Supplementary Fig. 4A)[16]. BM cell transplantation after irradiation allows hematopoietic reconstitution, including perivascular LYVE-1+ tissue-resident macrophages (Supplementary Fig. 5). *JAK2V617F* mutation only in vascular cells did not induce myeloproliferative neoplasm (Supplementary Fig. 4B) and did not induce elastin degradation (Supplementary Fig. 4D) or aortic dilation (Supplementary Fig. 4C). Moreover, no death was recorded during the 3-month follow-up in the 2 groups of chimeric mice (Supplementary Fig. 4E).

In order to investigate the role of *JAK2V617F* mutation specifically in endothelial cells in the onset of spontaneous dissecting aortopathy, we next generated Jak2V617F^WT/Flex VE-cadherin-Cre-ERT2 (herein referred to as *Jak2^V617F EC*) mice. We crossed Jak2V617F^WT/Flex mice with inducible VE-cadherin-Cre-ERT2 mice expressing the Cre recombinase after tamoxifen injection only in endothelial cells (Supplementary Fig. 6A). The absence of hematopoietic recombination in this model, as well as adequate endothelial recombination has been previously validated[16,18]. As expected, mice did not develop myeloproliferative neoplasm (Supplementary Fig. 6B). Aortic wall structure was preserved in mice expressing *JAK2V617F* mutation in endothelial cells (Supplementary Fig. 6D). In *Jak2^V617F EC* mice, we observed neither aortic dilation (Supplementary Fig. 6C) nor premature death (Supplementary Fig. 6E). Altogether, these experiments indicate that *JAK2V617F* mutation in vascular cells was not responsible for dissecting aortopathy.

### JAK2V617F mutation specifically expressed in myeloid cells induces dissecting aortopathy

To assess the implication of *JAK2V617F* mutation in hematopoietic cells, we generated chimeric mice expressing *JAK2V617F* mutation only in hematopoietic cells by transplanting BM cells from *Jak2^V617F HC-EC* mice into lethally irradiated C57BL/6 mice. Irradiated C57BL/6 mice transplanted with C57BL/6 BM cells were used as controls (Fig. 2A). Hematopoietic expression of *JAK2V617F* induced a myeloproliferative neoplasm (Fig. 2B), elastin degradation (Fig. 2C) and spontaneous aortic dilation at 1 (Supplementary Fig. 7A) and 3 months (Fig. 2D) after

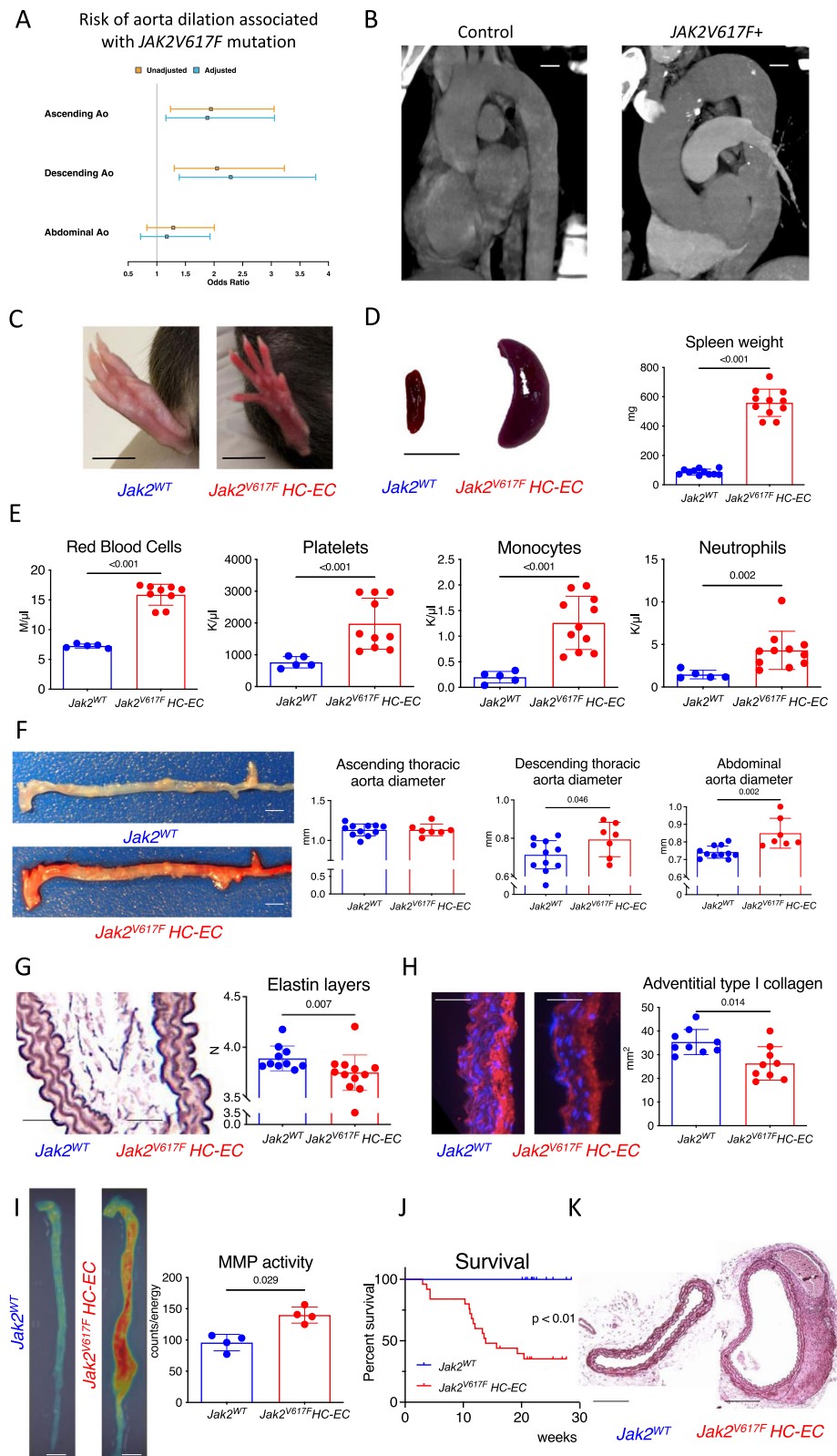

BMT. In addition, irradiated *C57BL/6* mice were transplanted with 100% *Jak2^WT* BM cells or a mixture of 80% *Jak2^WT*/20% *Jak2^V617F* HC-EC BM cells. One month after BMT, we observed aorta dilation in chimeric mice receiving 20% *Jak2^V617F* BM cells, despite no major changes in blood cell subsets (Supplementary Fig. 7B, C).

Next, to evaluate the implication of *JAK2V617F* in myeloid cells, we crossed *Jak2V617F^WT/Flex* mice with *LysMCre* ± mice expressing the Cre recombinase under Lysozyme M promoter, mainly expressed in myeloid cells (Supplementary Fig. 7D). *Jak2V617F^WT/Flex LysMCre^+/−*, referred to as *Jak2^V617F* Myel, developed myeloproliferative neoplasm, as attested by palmar erythema (Supplementary Fig. 7E), splenomegaly (Supplementary Fig. 7F), and higher number of circulating red blood cells, neutrophils and monocytes compared to littermate controls. However, platelet count was not different between groups (Fig. 2E).

**Fig. 1 | JAK2V167F mutation leads to aortopathy in both human and mice.**
**A** Adjusted risk factor for aorta dilation in a cohort of *JAK2V167F* + patients (*n* = 157) and age-gender-matched controls (*N* = 157), scale bar 2 cm. **B** CT-scan 3D-reconstruction of *JAK2V167F* + patient (right) and control (left) showing ascending and descending thoracic aorta. **C** Paw pictures of 7-week-old *Jak2^WT^* control *and Jak2^V617F^ HC-EC* mice showing palmar erythema in mutant mice, scale bar 1 cm.
**D** Quantitative analysis and representative photomicrographs of spleen size in 7-week-old control *Jak2^WT^(N = 11) and Jak2^V617F^ HC-EC* mice (*N* = 11) (scale bar 1 cm).
**E** red blood cell, platelet, and myeloid cell count of 7-week-old control *Jak2^WT^(N = 5)* and *Jak2^V617F^ HC-EC* mice (*N* = 9). **F** Representative photomicrographs and quantitative analysis of the mean aortic diameter in 7-week-old control *Jak2^WT^(N = 11)* and *Jak2^V617F^ HC-EC* mice (*N* = 7) in the thoracic (ascending and descending) and the abdominal aorta (scale bar 1 mm). **G** Quantification of the number of elastin layers

in the aortic wall by orcein staining in 7-week-old animals (*N* = 10–12/group), scale bar 50 μm. **H** Quantification of the collagen content (Cola1 immunostaining) in the aortic wall in 7-week-old animals (N = 9/group), scale bar 50 μm. **I** Fluorescent Molecular Tomography quantification of matrix metalloproteinase (MMP-2, −3, −9 and −13) activity in aorta of 10-week-old animals (*N* = 4/group), scale bar 1 mm. **J** Survival curve of control *Jak2^WT^* (*N* = 15) and *Jak2^V617F^ HC-EC* mice (*N* = 25).
**K** Representative photomicrographs of descending thoracic aorta after Orcein staining showing dissecting aortic aneurysm in *Jak2^V617F^ HC-EC* mouse (representative of 10 aortas/group), scale bar 100 μm. Data are presented as mean values ± SD. [Two-tailed Mann-Whitney test, *\*P* < 0.05, *\*\*P* < 0.01, *\*\*\*P* < 0.001]. Difference in survival was evaluated using log-rank test. *CT* computed tomography, *HC-EC* hematopoietic cells-endothelial cells, *MMP* matrix metalloprotease. Source data are provided as a Source Data file.

Histological analysis of aorta from 7-week-old *Jak2^V617F^ Myel* mice phenocopied the vascular phenotype of *Jak2^V617F^ HC-EC* mice. Indeed, *JAK2V617F* gain-of-function mutation under *LysM* promoter led to elastin degradation, as assessed by the significantly lower number of elastin layers in the abdominal aorta of *Jak2^V617F^ Myel* mice compared with the aorta of control *Jak2^WT^ Myel* mice (4.5 ± 0.24 *vs* 4.2 ± 0.09 layers, *P* < 0.05) (Supplementary Fig. 7G), which was associated with a reduction in collagen type 1a content (Fig. 2F, G) but no difference in α-SMA + smooth muscle cells content (Supplementary Fig. 7H). Global MMP activity in the aortic wall of *Jak2^V617F^ Myel* mice was significantly increased (Fig. 2H, I). At 7 weeks of age, we observed dilation in ascending thoracic (0.90 ± 0.06 *vs* 1.12 ± 0.05 mm, *P* < 0.05), descending thoracic (0.66 ± 0.12 *vs* 0.86 ± 0.10 mm, *P* < 0.05) and abdominal (0.72 ± 0.12 *vs* 0.89 ± 0.11 mm, *P* < 0.05) aorta in *Jak2^V617F^ Myel* mice, compared to *Jak2^WT^ Myel* mice (Fig. 2J). Finally, we observed that all *Jak2^V617F^ Myel* mice (100%) spontaneously died before 15 weeks of age. Autopsy was systematically performed and confirmed that deaths were due to dissecting aortic aneurysm, mainly associated with thoracic and/or abdominal hemorrhage (Fig. 2K). Taken together, these findings demonstrate that the presence of the *JAK2V617F* mutation in myeloid cells was responsible for the development of dissecting aortic aneurysm.

In order to understand the mechanisms underlying the lethal *JAK2V617F*-dependent aortic disease, we performed RNA sequencing of the aorta of 7-week-old *Jak2^V617F^ HC-EC* and control *Jak2^WT^* mice. The whole transcriptomic profile of the aorta was markedly altered by *JAK2V617F* global mutation, with upregulation of the expression of 137 genes and downregulation of the expression of 70 genes (Fig. 3A). Gene Ontology analysis showed that inflammatory pathways were upregulated in *Jak2^V617F^ HC-EC* aorta, including those associated with myeloid cell differentiation, homeostasis and development (Fig. 3B). Based on this finding, we investigated the consequences of *JAK2V617F* mutation on the local inflammatory response. First, we analyzed the immune cell composition in aorta of 7-week-old *Jak2^V617F^ HC-EC* mice and controls by flow cytometry, after enzymatic digestion. We did not find any significant differences in monocyte and neutrophil counts (Fig. 3C & Supplementary Fig. 8A), but we observed a 3-fold increase in CD45^+^CD11b^+^Ly6G^-^F4/80^high^CD68^high^ macrophages in the aorta from *Jak2^V617F^ HC-EC* mice (*P* < 0.05) (Fig. 3D). This result was confirmed by immunostaining in *Jak2^V617F^ HC-EC* (Fig. 3E) and *Jak2^V617F^ Myel* mice (Supplementary Fig. 8B). The significant increase in *Il1β, Il6* and *Tnfα* mRNA expression in aorta of *Jak2^V617F^ HC-EC* mice supports a vascular pro-inflammatory immune response in mutant mice (Fig. 3F). Notably, most of vascular CD68^+^ macrophages in mutant mice with aortic dilation expressed LYVE-1 (Fig. 3G) and a significant number expressed Ki-67, a proliferation marker (Supplementary Fig. 8C). However, at more advanced stage, most of CD68^+^ macrophages in *Jak2^V617F^ HC-EC* mice with severe aortic aneurysm did not colocalize with LYVE-1^+^ staining (Fig. 3G). All these observations suggest that macrophage accumulation in mutant mice at early stages unlikely resulted from the differentiation of infiltrating blood monocytes, but most likely from an

expansion of tissue-resident macrophages. Transcriptomic analysis confirmed that genes related to the phenotype of tissue-resident macrophages (*F13a1, Lyve1, Pf4,* and *Gas6*)[19] were enriched in the aorta of *Jak2^V617F^ HC-EC* mice (Fig. 3H).

## Jak2V617F Myel aortas harbor pro-inflammatory Lyve1 + resident macrophages

To better characterize aortic macrophage populations in *Jak2^V617F^ Myel* mice, we performed cellular indexing of transcriptomes and epitopes by sequencing (CITE-seq)[20] of total CD45^+^ cells from *Jak2^WT^* (*n* = 5) and *Jak2^V617F^ Myel* (*n* = 5) aortas (Fig. 4A). At sacrifice, among *Jak2^V617F^ Myel* mice, macroscopically observable aortopathy was seen in all mice, 2 mice displayed aortic aneurysm and 3 mice exhibited aortic dilation. Based on the expression of canonical cell surface markers and lineage-specific transcripts, immune cell subsets were identified and annotated (Fig. 4B, Supplementary Fig. 9A, B). Cells corresponding to monocytes/macrophages were extracted and reclustered separately to better resolve discrete subpopulations (Fig. 4A–C). We recovered several populations of LYVE-1^+^ tissue-resident macrophages (*Lyve1, Mrc1*; surface CD163, TIMD4, CD169, Fig. 4D) and recruited monocytes/macrophages (*Ccr2*; surface CD14, Fig. 4D). Differential gene expression analysis revealed three distinct populations of LYVE-1^+^ tissue-resident macrophages: Res-MGL1^hi^ (cluster A, enriched in *Clec10a* encoding MGL1), Res-*Txnip*^hi^ (cluster B: *Txnip, Trf, Stab1*) and Res-S100^hi^ (cluster C enriched in S100-protein encoding genes such as *S100a4, S100a6* or *S100a10*) (Fig. 4C–E). *Ccr2*^+^ inflammatory macrophages (cluster D) were enriched for e.g. *Spp1, Tnip3* or *Il1b* (Fig. 4C–E). Populations of type I IFN response macrophages (IFNIC, cluster E:[21] *Isg15, Ifit*), monocytes (cluster F: *Plac8, Ly6c2, Chil3*, surface Ly6C), and macrophages with high expression of early immediate-early genes[22] (cluster G, IEG + : *Egr1, Zfp36*) were also detected (Fig. 4C–E).

Distribution of macrophage subpopulations was dependent on genotype and disease severity. Pro-inflammatory *Ccr2*^+^ macrophages were particularly abundant in aortic aneurysm (59.6% of macrophages), while they were relatively rare in *Jak2^WT^* and *Jak2^V617F^ Myel* with limited aortic dilation (Fig. 4F, G), consistent with our observation of infiltration of Lyve1^-^ macrophages in the media of mutant mice with severe aortic aneurysm (Fig. 3G). The distribution of the different subsets of tissue-resident macrophages differed according to the genotype. Res-MGL1^hi^ were preferentially found in *Jak2^V617F^ Myel* aortas regardless of aortic disease severity (61.4% and 56.0% of tissue-resident macrophage in *Jak2^V617F^ Myel* aortas with aortic aneurysm and dilation, respectively), and Res-*Txnip*^hi^ were more abundant in *Jak2^WT^* aortas (60.2% of tissue-resident macrophages), whereas Res-S100^hi^ macrophages were evenly distributed across genotypes (Fig. 4G). Moreover, all resident macrophage populations shared expression of markers previously reported as associated with aortic resident macrophages (*Lyve1, Folr2, Cbr2, F13a1, Gas6*) (Fig. 4D and Supplementary Fig. 9)[7,19,23], but Res-MGL1^hi^ macrophages, particularly abundant in *Jak2^V617F^ Myel* aortas, showed higher expression of pro-inflammatory transcripts (*Nlrp3, Tnf, Tnfsf9, Ccl2, Cxcl1, Cxcl2, Cxcl10*) (Fig. 4I). Interestingly,

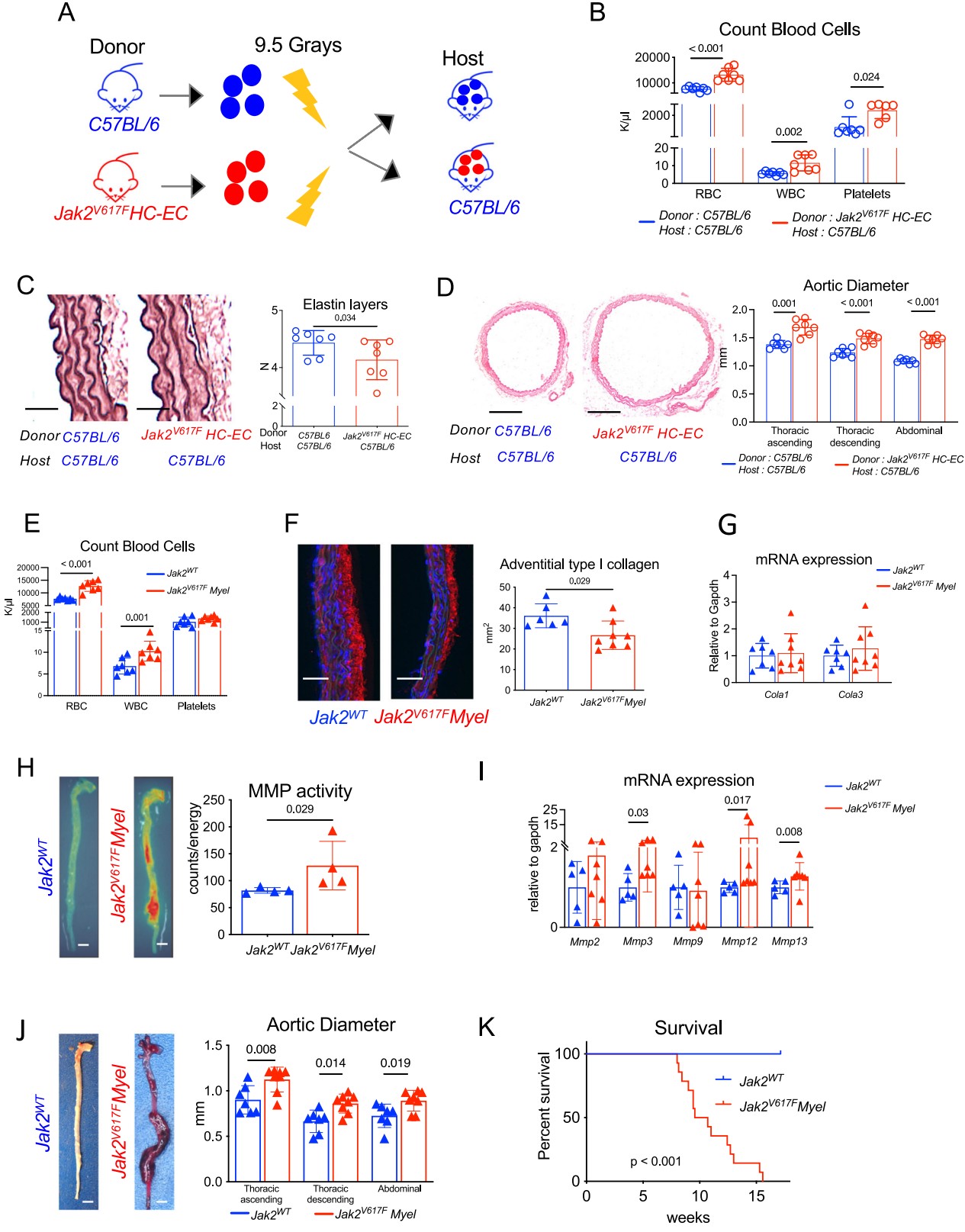

*Lyve1* expression was significantly increased in Res-MGL1^hi compared to Res-*Txnip*^hi macrophages, in agreement with our finding of higher expression levels of Lyve1 observed in bulk tissue transcriptome analysis of *Jak2*^V617F *HC-EC* aortas (Fig. 3H). Differential expression of cell surface markers measured by CITE-seq indicated that besides MGL1, Res-MGL1^hi macrophages exhibited higher surface levels of pro-inflammatory proteins such as CD95 (Fas), CD178 (Fas-Ligand), CD40

or JAML (Fig. 4J). All these data confirmed a pro-inflammatory signature in the aorta of *Jak2*^V617F *Myel* mice but pathways related to angiogenesis and lymphoangiogenesis were not impaired in mutant aortas (Supplementary Fig. 10A, B). Altogether, these results indicate that *JAK2V617F* mutation in myeloid cells not only increased the number of aortic resident Lyve-1^+ macrophages, but also polarized them towards a pro-inflammatory pathogenic phenotype.

**Fig. 2 | JAK2V167F mutation in hematopoietic cells promoted dissecting aortopathy. A** Experimental protocol of WT or *Jak2^{V617F}* HC-EC bone marrow cell transplantation in lethally irradiated *C57BL/6* mice (7/Group). **B** Red blood cell, white blood cell and platelet count of 20-week-old chimeric mice (*N* = 7/group). **C** Quantification of the number of elastin layers in the aortic wall by Orcein staining at sacrifice (*N* = 7/group), scale bar 50 µm. **D** Representative photomicrographs and quantitative analysis of the mean aortic diameter in WT or *Jak2^{V617F}* HC-EC chimeric mice (*N* = 7/group) in the thoracic (ascending and descending) and the abdominal aorta, scale bar 0.5 mm. **E** Red blood cell, white blood cell and platelet count of 7-week-old *Jak2^{WT} and Jak2^{V617F} Myel* mice (*N* = 7/group). **F** Quantification of the collagen content (Cola1 immunostaining) in the aortic wall in 7-week-old *Jak2^{WT}*(*N* = 6) and *Jak2^{V617F} Myel* animals (*N* = 8), scale bar 50 µm. **G** Quantification of *Cola1* and *Cola3* mRNA expression in aorta of 7-week-old *Jak2^{WT}* (*N* = 7) and *Jak2^{V617F} Myel* mice

(*N* = 8). **H** Fluorescent Molecular Tomography quantification of matrix metalloproteinase (MMP-2, −3, −9, and −13) activity in aorta of 9-week old *Jak2^{WT}* and *Jak2^{V617F} Myel* mice (*N* = 4/group), scale bar 1 mm. I, quantification of *Mmp2, Mmp3, Mmp9, Mmp12, Mmp13* mRNA expression in aorta of 7-week-old *Jak2^{WT}* (*N* = 5) and *Jak2^{V617F} Myel* mice (*N* = 7). **J** Representative photomicrographs and quantitative analysis of the mean aortic diameter in 7-week-old *Jak2^{WT}* (*N* = 7) and *Jak2^{V617F} Myel* (*N* = 8) mice in the thoracic (ascending and descending) and the abdominal aorta, scale bar 1 mm. **K** Survival curve of *Jak2^{WT}* (*N* = 11) and *Jak2^{V617F} Myel* mice (*N* = 14). Data are presented as mean values ± SD. [Two-tailed Mann-Whitney test, \*P < 0.05, \*\*P < 0.01, \*\*\*P < 0.001]. Difference in survival was evaluated using log-rank test. HC-EC, hematopoietic cells-endothelial cells; MMP, matrix metalloprotease. Source data are provided as a Source Data file.

## The JAK2V617F mutation in vascular tissue-resident macrophages induces dissecting aortic aneurysm

We then sought to generate a genetic model in which the *JAK2V617F* mutation would be preferentially induced in the tissue-resident macrophages. Our selection of an adequate Cre-driver was guided by data obtained from scRNA-seq analysis. Previous scRNA-seq studies have identified Platelet factor 4 (*Pf4*), highly expressed in megakaryocytes, as a marker of aortic resident macrophages[19,24], which we confirmed using an integrated dataset combining several independent single-cell studies of healthy and diseased mouse aortas[25] (Fig. 5A). Likewise, in our newly generated data, *Pf4* was highly expressed in LYVE-1⁺ resident macrophage subsets, but at much lower levels in CCR2⁺ inflammatory macrophages and monocytes (Fig. 5B). In scRNA-seq data from mouse blood CD11b⁺ cells[26], *Pf4* expression was absent from monocytes, but readily detected in contaminating platelets (Supplementary Fig. 11A). In line with our scRNA-seq analysis, previous studies using fluorescent reporter have demonstrated efficient *Pf4*-Cre-induced recombination in CD206⁺LYVE-1⁺ perivascular macrophages[24], but limited recombination in circulating monocytes[27]. Based on these findings, we crossed Jak2V617F ^{WT/Flex} mice with Pf4 ^{Cre+/-} mice expressing the Cre recombinase under Pf4 promoter, to generate transgenic mice with a preferential recombination of *JAK2V617F* mutation in tissue-resident macrophages (Fig. 5C). *Jak2V617F^{Flex}Pf4^{Cre+/-}* mice, referred to as *Jak2V617F Pf4*, developed aortic dilation and aneurysm before red blood cell and white blood cell increased in the blood. Indeed, at 7 weeks of age, red blood cell and white blood cell counts in the blood were not different between *Jak2^{V617F} Pf4* and *Jak2^{WT}* controls (Fig. 5D), but the aorta was significantly dilated in *Jak2^{V617F} Pf4* mice (Fig. 5E, F), associated with reduced peri-adventitial collagen 1a content (Fig. 5G, H), increased MMP signature (Fig. 5I) and marked accumulation of LYVE-1⁺ macrophages (Fig. 5J). The significant increase in *Il6 and Tnfα* mRNA expression in aorta of *Jak2^{V617F} Pf4* mice supports a vascular pro-inflammatory immune response in mutant mice (Fig. 5K). During the follow-up, 89% (34/38) of *Jak2^{V617F} Pf4* mice spontaneously died of dissecting aortic aneurysm, whereas only one death was observed in the control group (1/28, 4%) (*P* < 0.0001) (Fig. 5L).

## Pharmacological activation of vascular tissue-resident macrophage activity induces dissecting aortic aneurysm in C57BL/6 mice

Finally, we aimed to test a pharmacological approach in immunocompetent animals to activate in vivo vascular resident macrophages. Using an integrated dataset combining several independent single-cell studies of healthy[19,23] (Supplementary Fig. 12A) and diseased mouse aortas (Supplementary Fig. 12B), we found that the erythropoietin receptor (EpoR) was highly expressed in aortic resident macrophages. Flow cytometry analysis in different compartments confirmed that vascular resident macrophages highly and selectively expressed EPOR (Supplementary Fig. 12C), but circulating monocytes did not. Based on these observations, we hypothesized that EPO supplementation might activate vascular tissue-resident macrophages. Eight-week-old *C57BL/6* mice were treated intraperitoneally with EPO or PBS, and histological

analysis of the aorta were performed after 4 weeks of treatment and survival was recorded during 7 weeks of supplementation (Supplementary Fig. 12D). EPO treatment induced palmar erythema and polycythemia, but did not impact blood leukocyte and platelet counts (Supplementary Fig. 12E, F). EPO treatment induced a similar vascular phenotype to that seen in *JAK2V617F* mutant mice, with elastin degradation (Supplementary Fig. 12G), thinning of adventitial collagen layer (Supplementary Fig. 12H, I), local increase in proliferating resident macrophages (Supplementary Fig. 12J, K) and enhanced vascular pro-inflammatory responses (Supplementary Fig. 12L, M). EPO treatment promoted aortic dilation after 4 weeks and 7 weeks (Supplementary Fig. 12N, O), as well as aortic aneurysm (0 vs 58 %, *P* < 0.05) and lethal rupture (0 vs 42%, *P* < 0.05) (Supplementary Fig. 12P).

## Pharmacological depletion of vascular tissue-resident macrophage or JAK/STAT pathway blockade attenuates aortic disease severity in JAK2V617F mutant mice

Taken together, our findings provided strong evidence that *JAK2V617F* gain-of-function mutation in vascular tissue-resident macrophages promoted dissecting aortic aneurysm. We sought to further evaluate the effect of ablating aortic resident macrophages in the vascular wall on disease severity using Ki20227, an inhibitor of macrophage colony stimulating factor 1 (CSF1) receptor tyrosine kinase. In agreement with a previous report by Lim et al.[28], we confirmed that Ki20227 treatment (Fig. 6A) markedly depleted aortic macrophages (Fig. 6B, C), but did not affect red blood cells, white blood cells and platelets in blood (Fig. 6D). To assess the role of aortic resident macrophage depletion on aortic disease development, Ki20227 treatment was orally administered in *Jak2^{WT}* and *Jak2^{V617F} Myel* mice alternatively (Fig. 6E) to limit long term hematopoietic toxicity (Supplementary Fig. 13A−C). Ki20227 treatment remarkably abolished lethal aorta rupture in *Jak2^{V617F} Myel* mice (Fig. 6F). Histological analysis performed at 15 weeks in surviving Ki20227-treated *Jak2^{WT}* and *Jak2^{V617F} Myel* mice showed that depletion of vascular tissue-resident macrophages (Fig. 6G) abolished the pro-inflammatory signature in the mutant aorta (Fig. 6H) and alleviated both *JAKV617F*-mediated aortic remodeling (Supplementary Fig. 13D) and dilation (Fig. 6I).

Finally, we tested a pharmacological approach with ruxolitinib, a JAK2 inhibitor, currently used to treat *JAK2V617F* + patients with MPN. *Jak2^{V617F} HC-EC* and *Myel mice* were treated with PBS or ruxolitinib (oral gavage) daily during 5 and 10 weeks, respectively (Fig. 6J, M). Ruxolitinib attenuated the pro-inflammatory signature of aortic wall cells with decreased mRNA expression of *Il1β, Il6, Tnfα* and *Ccl2* in the aorta of ruxolitinib-treated *Jak2^{V617F} HC-EC* mice, compared to PBS-treated *Jak2^{V617F} HC-EC* mice (Fig. 6K). Ruxolitinib significantly limited thoracic aorta dilation (Fig. 6L) and decreased lethal aortic aneurysm rupture (Fig. 6N).

## Discussion

Here, we identified the *JAK2V617F* gain-of-function mutation as an independent risk factor for abnormal dilation of the ascending and

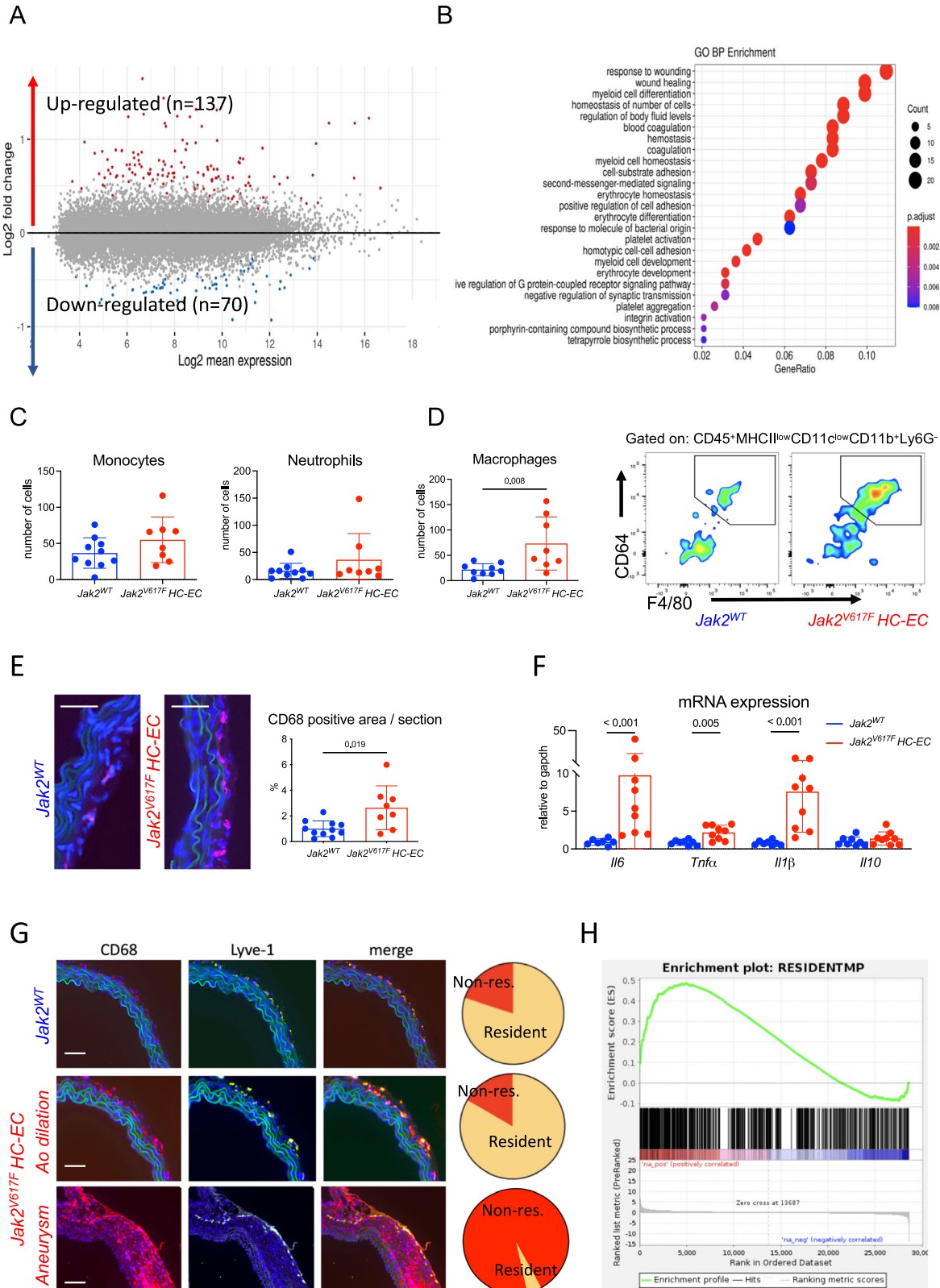

descending thoracic aorta. We showed that *JAK2V617F* in mice drives vascular tissue-resident macrophages toward a pro-inflammatory phenotype, responsible for aortic dilation and spontaneous dissecting aortic aneurysm.

Until now, little was known about the impact of *JAK2V617F* mutation on aortic aneurysm disease. Findings from a recently published study in angiotensin II-infused chimeric hypercholesterolemic *Apoe*[−/−] mice, which took place while we conducted our study, agree with ours regarding the association between the *JAK2V617F* mutation and aortic aneurysm[29]. Yet, their experimental model was not clinically relevant and did not recapitulate the human disease. Moreover, this study did not provide any mechanistic insights into how *JAK2V617F* mutation can promote aortopathy and, in particular, made no mention of the central role of resident macrophages in *JAK2V617F*-induced

**Fig. 3 | JAK2V167F mutation induces pro-inflammatory signature in the aortic wall. A** Difference in gene expression in the aorta of 7-week-old *Jak2^{WT}* (*N* = 4) and *Jak2^{V617F}* HC-EC mice (*N* = 3). **B** Top Gene Ontology terms mostly enriched among genes upregulated in the aorta of *Jak2^{V617F}* HC-EC mice when compared to *Jak2^{WT}* mice. **C** Flow cytometry quantification of CD45 + CD11c^{low}CMHII^{low}CD11b + Ly6G-F4/80^{low-}CD64^{low}Ly6C + monocytes and CD45 + CD11c^{low}CMHII^{low}CD11b + Ly6G+ neutrophils in the aorta of 7-week-old *Jak2^{WT}* (*N* = 10) and *Jak2^{V617F}* HC-EC animals (*N* = 8). **D** Flow quantification and representative dot plot of CD45 + CD11c^{low}CMHII^{low}CD11b + Ly6G-F4/80^{high}CD64^{high} macrophage count in the aorta of 7-week-old *Jak2^{WT}* (*N* = 10) and *Jak2^{V617F}* HC-EC animals (*N* = 8). **E** Quantification and representative photomicrographs of CD68 positive macrophages in the aorta of 7-week-old mice (*N* = 11 in control group

and *N* = 8 in *Jak2^{V617F}* HC-EC group), scale bar 50 μm. **F** Quantification of *Il6, Tnfα, Il1β,* and *Il10* mRNA expression by RT-qPCR in the aorta of 7-week-old *Jak2^{WT}* (*N* = 8) and *Jak2^{V617F}* HC-EC animals (*N* = 9). **G** Immunofluorescence staining in the aortic wall of *Jak2^{WT}* and *Jak2^{V617F}* HC-EC mice, DAPI (Blue), CD68 (Red), LYVE-1 (Green), scale bar 50 μm. resident macrophages were defined as CD68 + LYVE-1+ cells (*N* = 5/group). **H** GSEA for genes related to a gene set signature for tissue resident macrophage[19] in the aorta of 7-week-old *Jak2^{WT}* and *Jak2^{V617F}* HC-EC animals (*N* = 4/group). Data are presented as mean values ± SD. [Two-tailed Mann-Whitney test, *\*P* < 0.05, *\*\*P* < 0.01, *\*\*\*P* < 0.001]. HC-EC, hematopoietic cells-endothelial cells; *Il*, interleukin; *Tnf*, tumor necrosis factor; GSEA, gene set enrichment analysis. Source data are provided as a Source Data file.

severe aortopathy. In our study, using several complementary approaches, we showed that mice expressing human *JAK2V617F* gain-of-function mutation developed spontaneously aortic dilation at early stages and dissecting aortic aneurysm later on, in the absence of any challenge, and we identified the critical impact of *JAK2V617F* mutation on functions of vascular resident macrophages.

Histological analysis showed increased macrophage content in the aorta of young mutant mice. Early macrophage infiltration in the aortic wall is a hallmark of aortic wall development both in mice and in humans[30,31]. Macrophages largely participate in aortic wall degradation and remodeling through the production of chemokines, cytokines and proteases. In animal models using angiotensin II to promote aneurysm, macrophages detected in the aneurysmal aorta were shown to mostly derive from circulating monocytes[4,32], whereas in *Jak2^{V617F}* mutant mice, aortic macrophages unlikely derived from blood monocytes at early stages. At 7 weeks of age, *Jak2^{V617F}* mutant mice developed aortic dilation with no sign of infiltrating neutrophils or monocytes in the vascular wall, but with high numbers of proliferating macrophages expressing LYVE-1, a vascular tissue-resident marker. Global and single cell transcriptomic analysis confirmed that aortic macrophages in *Jak2^{V617F}* mutant mice exhibited a tissue-resident signature. Vascular tissue-resident Lyve-1+ macrophages have been shown to display protective functions against deleterious aorta remodeling in immunocompetent *C57BL/6* mice[28]. In our study, we found that tissue-resident Res-MGL1^{hi} macrophages, abundant in *Jak2^{V617F}* Myel aortas, expressed high levels of chemokine transcripts such as *Ccl2, Cxcl1, Cxcl2, Cxcl10*, as well as high levels of *Nlrp3* transcripts, suggesting that *JAK2V617F* mutation induced a shift in macrophage profile toward a pathogenic pro-inflammatory phenotype. Increased *Il1b, Il6* and *Tnfα* mRNA expression, as well as increased MMP activity in the aorta of 8-week-old *Jak2^{V617F}* mutant mice, likely resulted from higher content of activated macrophages in the aorta. At more advanced stages, in *Jak2^{V617F}* mutant mice with large aortic aneurysm, another dominant subset of macrophages expressing *Ccr2*, but not *Lyve1*, has been identified in the aortic wall. These Ccr2+ Lyve-1- macrophages most likely derived from recruited blood monocytes in response to local production of chemokines, mainly Ccl2. Ccr2 is highly expressed on classical circulating monocytes and regulate their trafficking. Previous works have reported that in response to Angiotensin II infusion *Ccr2*-deficient mice are characterized by decreased infiltrating monocytes in the aortic wall, decreased local inflammation and finally less severe aortic disease[33,34].

The *JAK2V617F* mutation has been shown to occur in the majority of patients with polycythemia vera[35]. The mutation also induces a polycythemia in different mouse models, due to either the proliferation and survival of erythroid progenitors carrying the mutation or local erythropoiesis in the bone marrow resulting from the release of growth factors by mutant myeloid progenitors[36]. Indirect stimulation of erythropoiesis most likely accounted for polycythemia in *Jak2^{V617F}* Myel mice since the Jak2 mutation was not detected in erythroid progenitors. Polycythemia by itself might promote arterial dilation and decrease blood pressure through the activation of the endothelial NO synthase[37]. However, we can rule out the possibility that the Jak2-

associated polycythemia was responsible for aortic dilation in mutant mice since aortopathy was observed in *Jak2^{V617F}* Pf4 mice before polycythemia developed. In addition, disease severity was not affected by pharmacological correction of polyglobulia.

We also found that high doses of EPO induced dissecting aortic aneurysm in immunocompetent *C57BL/6* mice, supporting the critical role of the JAK2 pathway in aortic disease, but we did not specifically investigated which cell type was activated in response to EPO supplementation. Our results are in line with a recent study showing that EPO repetitive injection promoted aortic aneurysm in *Apoe^{-/-}* mice[38]. The authors found higher pro-inflammatory signature in the aorta of EPO-treated mice and suggested that endothelial cell activation might be involved in EPO-induced abdominal aortic aneurysm. However, in our study performed in normocholesterolemic mice, *JAK2V617F* gain-of-function mutation in vascular cells, and specifically in endothelial cells, did not promote aortic disease.

The relevance of our results in experimental murine models to the human situation are supported by the present finding of aortic dilation in patients with *JAK2V617F* mutation. In addition, several case reports also described arterial aneurysms in patients with JAK2+ myeloproliferative neoplasm[39,40]. More recently, in a genome-wide association study (GWAS) using the UK Biobank, an intronic variant located within the gene encoding JAK2 tyrosine kinase has been linked to aortic aneurysm[41]. Conversely to what we observed in *Jak2^{V617F}* mutant mice, dissecting aortic aneurysm is not common in *JAK2V617F+* patients. This discrepancy could be explained by the fact that *JAK2V617F* mutation in human is a somatic mutation occurring at late stages in adult whereas it is a germline mutation in mice who developed severe hematological disease at early stages of life. In addition, this discrepancy between human and mice data might be accounted for by the vascular protection by specific drugs, such as ruxolitinib, that are currently used to treat *JAK2V617F+* myeloproliferative diseases. Interestingly, in our study, we showed that ruxolitinib attenuated pro-inflammatory signature in the aortic wall and protected against deleterious aortic remodeling and lethal rupture.

The protection induced by Ki20227 treatment supports the role of resident vascular macrophages in the pathophysiology of JAK2V617F-mediated aortopathy. However, Ki20227 could have off-target effects, such as angiogenesis inhibition[42]. However, in our mutant models, we did not find any evidence that *JAK2V617F* mutation impaired angiogenesis in the aortic wall.

## Methods

### Human samples analysis

We screened in a cohort of *JAK2V617F* + patients treated in Saint-Antoine Hospital (Assistance Publique-Hôpitaux de Paris), those who had a thoraco-abdominal CT-scan during the last 5 years. Controls were gender- and age-matched patients hospitalized in non-hematology department (Saint-Antoine Hospital) and who had a thoraco-abdominal CT-scan during the same period. Cardiovascular risk factors were recorded and used for multivariate adjustment. Aorta measurements were performed by two radiologists worked on the original axial data sets and used electronic calipers to measure the

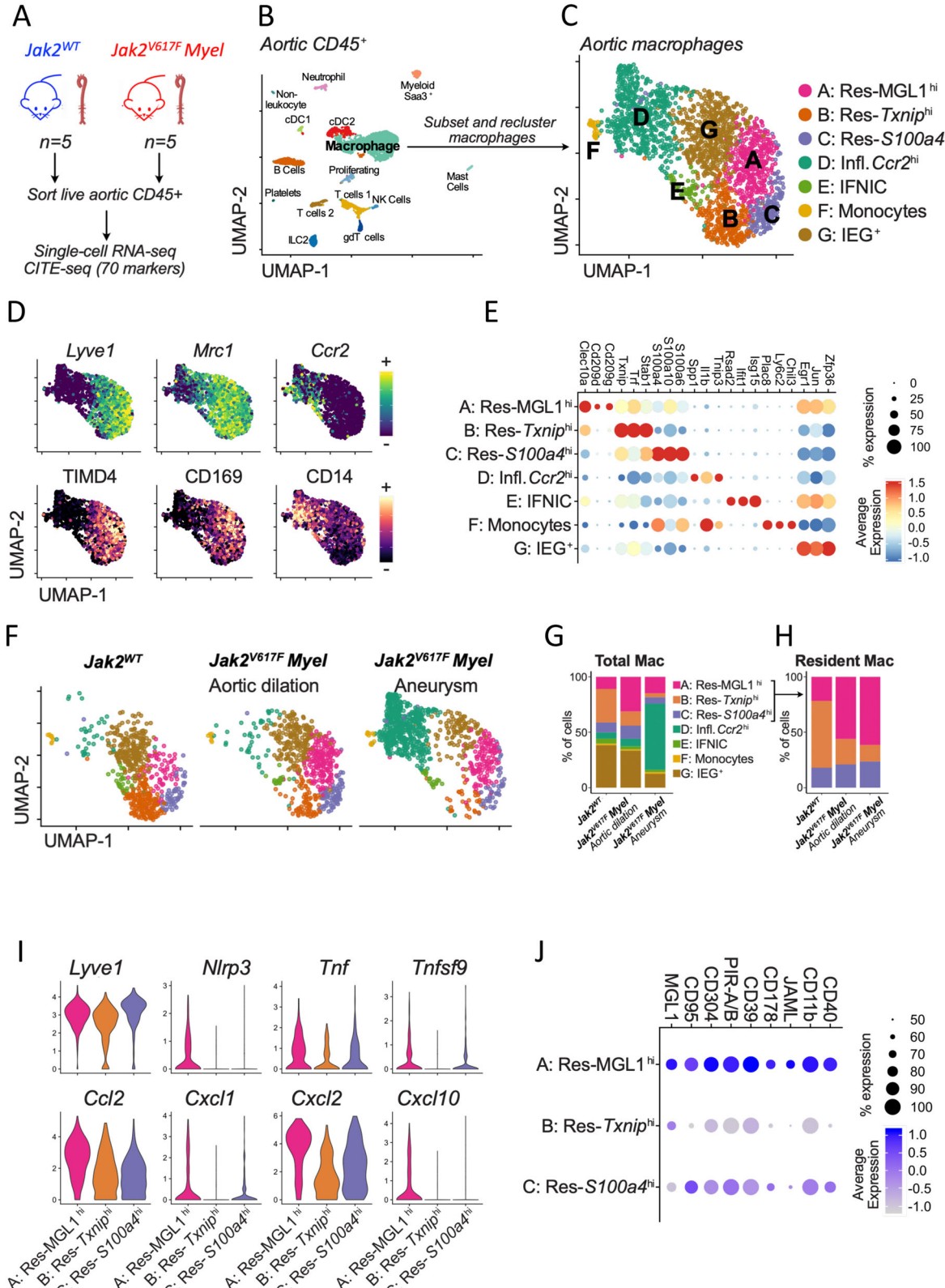

largest diameter of aorta at three points: (1) the mid ascending aorta at the level of the right pulmonary artery, (2) the mid descending aorta at the level of the aortic valve and (3) the maximal infrarenal aortic diameter. In order to evaluate the effect of the *JAK2V617F* mutation on the aorta diameter, we perfomed both univariate and multivariate logistic regression. Classical risk factors of aortic aneurysm such as age, arterial hypertension, diabetes, smoking status and gender was included in the multivariate logistic regression. The aorta was considered as dilated if diameter was higher than the average of the whole included patients.

Written informed consent was not required, in compliance with French law on retrospective studies of anonymized data (loi Jardé, MR004). All local medico-administrative data bases are declared to the general register of data processing of the Assistance publique-

**Fig. 4 | *Jak2^{V617F}* Myel aortas harbor pro-inflammatory Lyve1 + tissue resident macrophages. A** Experimental design, **B** UMAP visualization of scRNA-seq profiles of CD45 + cells extracted from *Jak2^{WT}* and *Jak2^{V617F}* Myel aortas. **C** Single-cells annotated as macrophages on the UMAP displayed in B were in silico extracted, reanalyzed with dimensional reduction (UMAP) and clustered identifying 7 major populations (Annotated A to G). **D** Expression intensity of transcripts (top) and cell surface markers (bottom) characteristic of tissue resident macrophages (Lyve1, Mrc1, TIMD4, CD169) and recruited monocytes/macrophages (Ccr2, CD14) projected on the macrophage UMAP plot. **E** Expression of selected marker transcripts in the 7 macrophage populations. **F** Macrophage UMAP plot split according to experimental condition (color code as in **B**). **G** Proportions of macrophage populations among total aortic macrophages. **H** Proportions of tissue resident macrophage subsets among total aortic tissue resident macrophages. **I** Normalized expression of the indicated transcripts in tissue resident macrophage subpopulations. **J** Average expression of the indicated cell surface markers (measured by CITE-seq) in tissue resident macrophage subpopulations. UMAP, Uniform Manifold Approximation and Projection for Dimension Reduction, *sc* single cell, *CITE-seq* Cellular Indexing of Transcriptomes and Epitopes by Sequencing.

Hopitaux de Paris (Bureau de la protection des données, France no. 20220106150715).

## Murine models

Experiments were conducted according to the guidelines formulated by the European Community for experimental animal use (L358-86/ 609EEC) and approved by the Ethical Committee of INSERM CEEA34 (MESR26963) and the French Ministry of Agriculture (agreement A75-15-32). Mouse breeding occurred in our animal facility in accordance with local recommendations. Animals were provided with food and water ad libitum. All the animals were maintained under identical standard conditions (housing, regular care and normal chow). Mice were maintained in isolated ventilated cages under specific pathogenfree conditions. Before euthanasia by cervical dislocation, animals were anesthetized with isoflurane (3% in oxygen).

Control mice were matched with littermates of the appropriate, age, sex, and genetic background to account for any variation in data. All mice were on a C57BL/6 background. *Jak2^{V617F}* HC-EC mice were obtained by crossing VE-cadherin-Cre[15] (Jackson Laboratory) transgenic mice with Jak2V617F Flex/WT mice[43]. *Jak2^{V617F}* HC-EC mice have been generated by PE Rautou's group[16]. *Jak2^{V617F}* EC mice were obtained by crossing VE-Cadherin-cre-ERT2 transgenic mice (Provided by R.H. Adams, Max Planck Institute for Molecular Biomedicine, Münster, Germany)[44] with Jak2V617F Flex/WT mice. *Jak2^{V617F}* Myel mice were obtained by crossing LysM-Cre transgenic mice (Jackson Laboratory) with Jak2V617F Flex/WT mice. Jak2^{V617F} Pf4 mice were obtained by crossing *Pf4*-Cre transgenic mice[45,46] with Jak2V617F Flex/WT. In all experiments, male and female mice were used. For induction of Cre recombinase expression in Jak2V617F Flex/WT; VE-Cadherin-creERT2 mice, mice were injected intraperitoneally with tamoxifen (Sigma, T5648), 1 mg/mouse/day for 5 consecutive days over 2 consecutive weeks (10 mg in total per mouse) between the age of 5 to 7 weeks. The Flex (Flip-Excision) strategy allows expression of a mutated gene in adulthood, in a temporal and tissue-specific manner[47].

## Bone marrow transplantation

Six to 8-week-old *C57BL/6 J* and Jak2^{V617F} HC-EC mice were subjected to medullar aplasia following sublethal total body irradiation (9.5 grays). At day 1, irradiated mice were repopulated intravenously with bone marrow cells (10 millions) isolated from femurs and tibias of age matched *Jak2^{V617F}* HC-EC and *C57BL/6 J* CD45.1 or CD45.2 control mice. Mice were followed-up during 1–3 months with repetitive blood cells analysis and aortic diameter measurements by echography.

## Blood cell count analysis

Blood was collected through the mandibular vein in a tube pre-coated with EDTA after transient anesthesia (Isoflurane). Blood counts analyses were performed using a Hemavet 950FS analyser (Drew scientific).

## Fluorescence molecular tomography (FMT) imaging

Twenty-four hours before sacrifice, *Jak2^{V617F}* HC-EC and *Jak2^{V617F}* Myel animals were anesthetized with isoflurane and received 150 µL intravenously of a fluorescent imaging probe MMPsense 680 (NEV 10126, PerkinElmer). The probe is optically silent in its unactivated state and becomes highly fluorescent following activation by MMPs including MMP-2, −3, −9, and −13. Images were acquired using a fluorescence molecular imaging system (FMT 2500TM, VisEnMedical).

## In vivo treatment

Erythropoietin (Eprex, Janssen) (EPO) was administered for 5 and 7 consecutive weeks by intraperitoneal injection to 8-week-old C57BL/6 males (5000 UI/kg, 3 times per week). EPO was prepared in PBS. Control mice received the same volume of vehicle (PBS). Ruxolitinib (Jakavi, Novartis) was administered for 5 consecutive weeks (30 mg/kg, 2 times per day) by oral gavage to *Jak2^{V617F}* HC-EC. Ruxolitinib was prepared from 10 mg commercial tablets in PEG300/5% dextrose mixed at a 1:3 ratio. Control mice were administered the same volume of vehicle (PEG300/5% dextrose). Five-week-old *C57BL/6* mice received daily oral administration of Ki20227[9] by food for 2 weeks. 5-week-old *Jak2^{V617F}* Myel mice were alternatively fed with Ki20227 and chow diet for 10 weeks. Phenylhydrazine (PHZ) was administered every 3 days by intra peritoneal injection to 6-week-old *Jak2^{V617F}* HC-EC mice (25 mg/kg body weight). PHZ was prepared in PBS and control mice received the same volume of vehicle.

## Histological analysis

The aorta was flushed with cold PBS through the left ventricle and removed from the root to the iliac bifurcation. Abdominal aorta was used for histologic studies. The suprarenal region of the abdominal aorta, subjected to aneurysmal development, was harvested and embedded in both cryostat and paraffin and 7-µm thick sections were used for histology and immunohistochemistry analysis. Sections were stained with Orcein (MMFrance) for detection of elastin layers. The mean number of elastin layers was quantified by a researcher blinded to the experimental protocol (4 measurements/section, 8 sections/ mouse). Sections were stained for macrophages using a rat anti– mouse CD68 primary antibody (MCA1957, clone FA-11, Bio Rad) revealed with an anti–rat Cy5 secondary antibody (712-175-153, Jackson ImmunoResearch). The percentage of cellular area positive for CD68 staining (macrophages) was quantified using Histolab software (Microvision) and the CD68 + area/total aortic wall area (media + adventitia) ratio was calculated. For resident macrophages, sections were also stained with a rabbit anti-mouse LYVE-1 (ab33682, polyclonal, Abcam) revealed with an anti–rabbit Cy3 secondary antibody (711-165-152, Jackson ImmunoResearch). Total macrophages (CD68 +) and resident macrophages (CD68 + LYVE-1 +) were manually counted (8 sections per mouse for each experiment). Proliferative macrophages were stained with a rabbit anti-mouse Ki67 (RM-9106-S, clone SP6, Epredia) after fixation and permeabilization (Triton 0.01%) and revealed with an anti–rabbit Cy3 secondary antibody (711-165-152, Jackson ImmunoResearch). Total macrophages (CD68 +) and proliferative macrophages (CD68 + Ki67 +) were manually counted. The media was stained with a rabbit anti-mouse Alpha Smooth muscle actin (ab5694, polyclonal, abcam) after fixation and permeabilization (Triton 0.01%) and revealed with an anti–rabbit Cy3 secondary antibody (711-165-152, Jackson ImmunoResearch). The quantification was obtained by calculating the aSMA positive area and the total media area using Histolab software (Microvision) and the aSMA+ area/total media ratio was calculated. The adventital collagen was stained with a

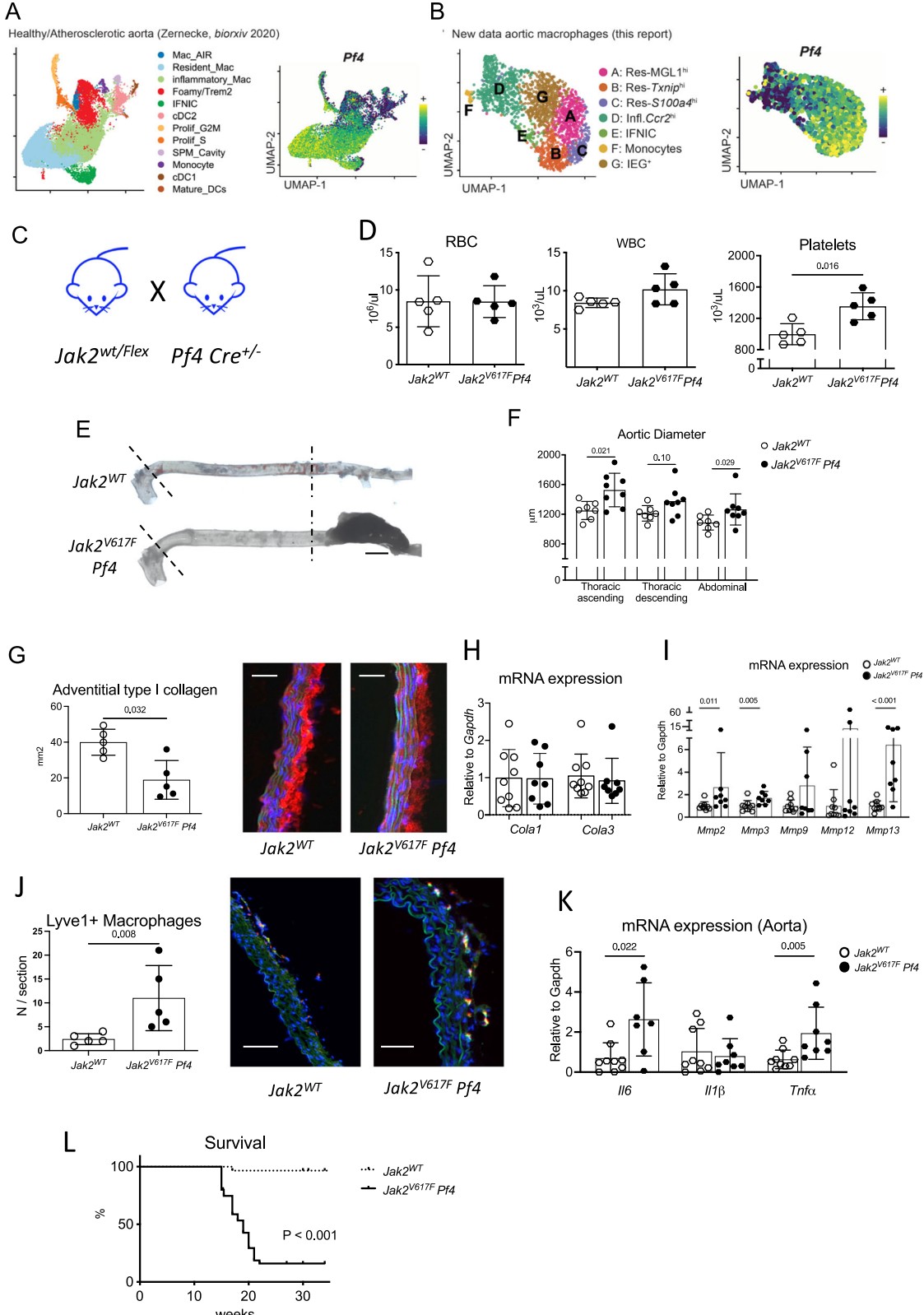

rabbit anti-mouse Collagen Type I (234167, polyclonal, Merck Milli-pore) revealed with an anti–rabbit Cy5 secondary antibody (711-175-152, Jackson ImmunoResearch). Total advential collagen positive area was calculated using ImageJ (NIH) software.

In each experiment, 8 sections per mouse were quantified. Histology quantifications were done by a researcher blinded to the experimental protocol. Epifluorescence analysis, brightfield

and polarized light imaging were done using Leica DM6000B microscope.

**Flow cytometry**

FACS analysis was performed on total aortas. The day of sacrifice, animals were perfused with cold PBS and whole aortas were rapidly cleaned and harvested under a dissection microscope. Aortas were

**Fig. 5 | JAK2 mutation in vascular tissue resident macrophage cells promotes dissecting aortopathy. A** Left, cells corresponding to monocytes/macrophages/ dendritic cells were extracted from murine healthy and atherosclerotic plaques and separately reanalyzed with dimensional reduction (UMAP)[25]. Right, expression of *Pf4* transcript projected on the vascular monocytes/macrophages/dendritic cells UMAP plot. **B** Left, cells corresponding to macrophages were extracted from murine aorta and separately reanalyzed with dimensional reduction (UMAP). Right, expression of *Pf4* transcript projected on the vascular tissue resident macrophage subsets UMAP plot. **C** *Jak2V617F WT/Flex* mice were backcrossed with *Pf4 Cre±* mice to generate control *Pf4 Cre- Jak2WT* and *Pf4 Cre+ Jak2V617F (Called Jak2V617F Pf4)* mice. **D** Red blood cell, white blood cell and platelet count of 7-week-old *Jak2WT and Jak2V617F Pf4* mice (*N* = 5/group). **E, F** representative photomicrographs and quantitative analysis of the mean aortic diameter in control 7-week-old *Jak2WT* (*N* = 7) and *Jak2V617F Pf4* mice (*N* = 8) in the thoracic (ascending and descending) and the

abdominal aorta, scale bar 1 mm. **G** Quantification of the collagen content (Cola1 immunostaining) in the aortic wall in 7-week-old *Jak2WT and Jak2V617F Pf4* animals (*N* = 5/group), scale bar 50 μm. **H** Quantification of *Cola1* and *Cola3* mRNA expression in aorta of 7-week-old *Jak2WT* (*N* = 8) and *Jak2V617F Pf4* mice (*N* = 9). **I** Quantification of *Mmp2, Mmp3, Mmp9, Mmp12, Mmp13* mRNA expression in aorta of 7-week-old *Jak2WT* (*N* = 8) and *Jak2V617F Pf4* mice (*N* = 9). **J** Quantification of LYVE-1+ CD68 + tissue resident macrophages in the aorta of 7-week-old mice (Immunostaining, *N* = 5/group), scale bar 50 μm. **K** Quantification of *Il6, Il1β* and *Tnfα* mRNA expression in aorta of *Jak2WT* (*N* = 8) and *Jak2V617F Pf4* mice (*N* = 9). **L** Survival curve (*N* > 30/group). Data are presented as mean values ± SD. [Two-tailed Mann-Whitney test, *\*P* < 0.05, *\*\*P* < 0.01]. Difference in survival was evaluated using log-rank test. *UMAP* Uniform Manifold Approximation and Projection for Dimension Reduction, *Pf* Platelet factor, *MMP* matrix metalloprotease, *Il* interleukin, *Tnf* tumor necrosis factor. Source data are provided as a Source Data file.

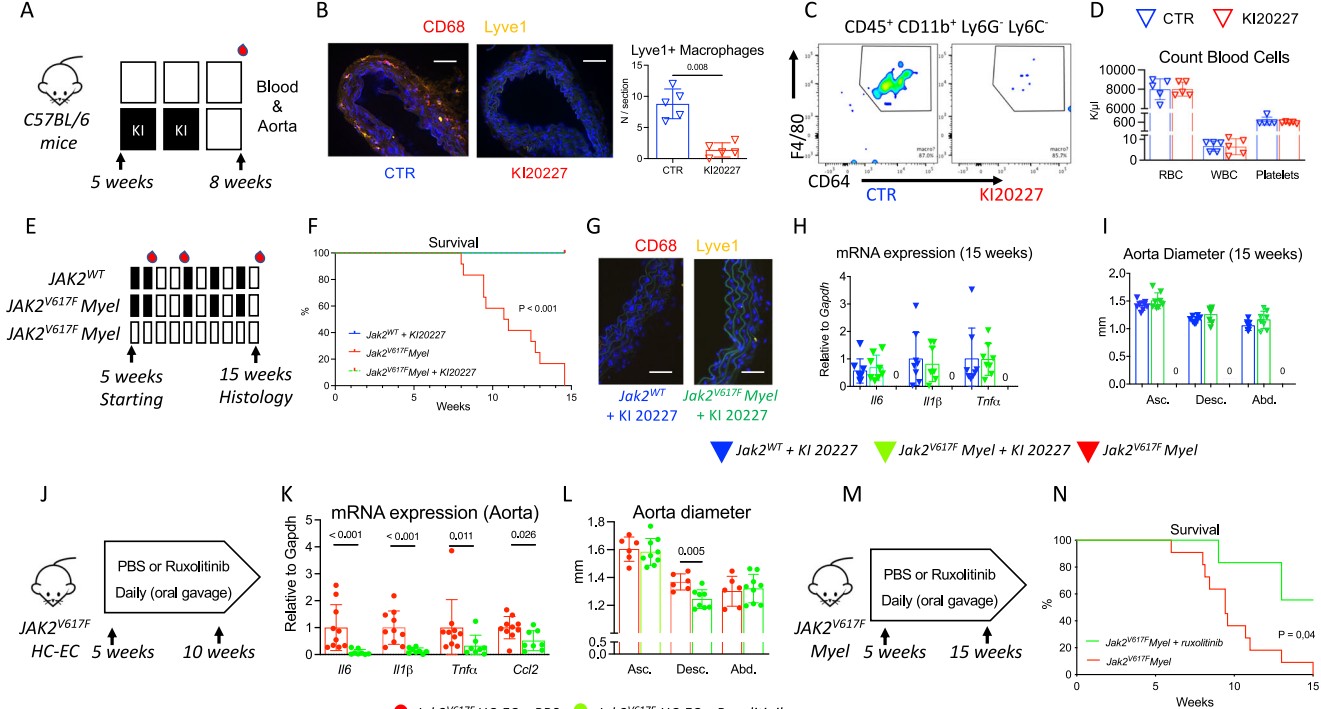

**Fig. 6 | Pharmacological blockade of vascular tissue resident macrophage cells attenuates *JAK2V617F*-mediated dissecting aortopathy. A** Experimental protocol of Ki20227 treatment in *C57BL/6* male mice. **B** Immunostaining of CD68 + LYVE-1+ macrophages (Yellow) in the aorta of Ki20227- or PBS-treated mice (*N* = 5/group), scale bar 50 μm. **C** Representative dot plot of CD45 + CD11b + Ly6G-Ly6C-CD64 + F4/80+ macrophage count in the aorta of Ki20227- or PBS-treated *C57BL/6* mice. **D** Red blood cell, white blood cell and platelet count of Ki20227- or PBS-treated *C57BL/6* mice. **E** experimental protocol of PBS or Ki20227 treatment in *Jak2WT* and *Jak2V617F Myel* mice. *Jak2WT* (blue, *N* = 8) and *Jak2V617F Myel* (Green, *N* = 8) mice received Ki20227 and a third group of *Jak2V617F Myel* (Red, *N* = 12) only received control diet without pharmacological drug. **F** survival curve of treated mice. **G** Immunostaining of CD68 + LYVE-1+ macrophages (Yellow) in the aorta of surviving Ki20227-treated *Jak2WT* (blue) and *Jak2V617F Myel* (green) mice (*N* = 8/group), scale bar 50 μm. **H** Quantification of *Il6, Tnfα, Il1β and Ccl2* mRNA expression by RT-qPCR in the aorta of surviving Ki20227-treated *Jak2WT* (blue) (*N* = 8) and *Jak2V617F Myel* (green) mice (*N* = 7). **I** Quantitative analysis of the mean aortic diameter of

surviving Ki20227-treated *Jak2WT* (blue) and *Jak2V617F Myel* (green) mice (*N* = 8/ group) in the thoracic ascending (Asc.), thoracic descending (Desc.) and the abdominal (Abd.) aorta. **J** Experimental protocol describing ruxolitinib (60 mg/kg, daily during 5 weeks) or PBS treatment in *Jak2V617F HC-EC* mice. **K** quantification of *Il6, Tnfα, Il1β and Ccl2* mRNA expression by RT-qPCR in the aorta of *Jak2V617F HC-EC* mice treated with PBS (Red) (*N* = 10) or Ruxolitinib (Green) (*N* = 8). **L** Quantitative analysis of the mean aortic diameter of PBS (Red, *N* = 7) or ruxolitinib (Green, *N* = 8) -treated *Jak2V617F HC-EC* mice using ultrasonography in the thoracic ascending (Asc.), thoracic descending (Desc.) and the abdominal (Abd.) aorta. **M** experimental protocol describing Ruxolitinib (60 mg/kg, daily during 10 weeks) or PBS treatment in *Jak2V617F Myel* mice. **N** survival curve of treated mice (*N* = 11 PBS and *N* = 7 Ruxolitinib). Data are presented as mean values ± SD. [Two-tailed Mann-Whitney test or Kruskal-Wallis **H, I**, *\*P* < 0.05, *\*\*P* < 0.01, *\*\*\*P* < 0.001]. Difference in survival was evaluated using log-rank test. HC-EC, hematopoietic cells-endothelial cells; *Il*, interleukin; *Tnf*, tumor necrosis factor. Source data are provided as a Source Data file.

minced and incubated 30 min at 37 °C in an enzymatic digestion cocktail (Collagenase I (450 U/ml, Sigma Aldrich), Collagenase XI (125 U/ml, Sigma Aldrich), Hyaluronidase (60 U/ml, Sigma Aldrich), DNAse (60 U/ml, Sigma Aldrich) and Elastase (0.372 U/ml, Sigma Aldrich)). Then a mechanic digestion was performed to pass the cells through a 100 μm filter before the staining. Leukocyte suspension was obtained from aorta and extracellular antigens were stained with

fluorescent labeled anti-mouse antibodies for 30 min at 4 °C. Flow cytometric acquisitions were done on a BD LSRFortessa (BD Biosciences) and data were analyzed using FlowJo Software (TreeStar, Inc.). Forward scatter (FSC) and side scatter (SSC) were used to gate live cells, red blood cells, debris, cell aggregates and doublets being excluded. Lymphocytes T were identified as CD45 + /CD11b-/ CD3 + ; Lymphocytes B as CD45 + /CD11b-/B220 + ; Neutrophils as

CD45 + /CD11b + /Ly6G + ; Monocytes as CD45 + /CD11b + /Ly6G-/Ly6C + ; Macrophages as CD45 + /CD11b + /Ly6G-/F4/80^high/CD64^high; Resident macrophages as CD45 + /CD11b + /Ly6G-/F4/80^high/CD64^high/Lyve-1 + ; Dendritic cells were CD45 + /CD11c^high/CMHII^high.

The following primary conjugated antibodies were used for staining in the aorta: anti-CD45 (Alexa Fluor 700, clone 30-F11, BD Biosciences), anti-CD11b (BV605, clone M1/70, BD Biosciences), anti-Ly6C (APC-Cy7, clone AL-21, BD Biosciences), anti-Ly6G (BUV395, clone 1A8, BD Biosciences), anti-CD64 (BV421, clone X54-5/7.1 Biolegend), anti-F4/80 (PE, clone CI:A3-1, Abcam), anti-CD11c (PerCP-Cy5.5, clone HL3, BD Biosciences), anti-MHC-II (BV510, clone M5/114.115.2, BD Biosciences), anti-CD3 (PE-Cy7, clone 145-2C11, BD Biosciences), anti-CD45R (B220) (BUV496, clone RA3-6B2, BD Biosciences), anti-LYVE1 (clone ALY7, Life Technologies), anti-EPO-R (PE, polyclonal, Cliniscience), anti-CD45.1 (BV605, clone A20, Biolegend), anti-CD45.2 (BV510, clone 104, Biolegend).

The following primary conjugated antibodies were used for the staining in the peritoneal macrophages and in the bone marrow: anti-CD45 (Alexa Fluor 700, clone 30-F11, BD Biosciences), anti-CD11b (BV605, clone M1/70, BD Biosciences), anti-Ly6C (APC-Cy7, clone AL-21, BD Biosciences), anti-Ly6G (BUV395, clone 1A8, BD Biosciences), anti-CD64 (BV421, clone X54-5/7.1 Biolegend), anti-F4/80 (FITC, clone BM8-1, Tonbo), anti-LYVE-1 (clone ALY7, Life Technologies), anti-EPO-R (PE, polyclonal, Cliniscience).

### Determination of JAK2V617F allele burden on bone marrow hematopoietic cells

Femur and tibia from control Jak2^WT and Jak2^V617F HC-EC mice were crushed and filtered in PBS + EDTA 2 mM+FBS 2%. Mature erythroid cells were then lysed in an ammonium buffer. For progenitor cells analysis, an enrichment in CD117 + (Kit + ) cells was performed by magnetic cell sorting using anti-CD117 microbeads according to manufacturer's instruction (Miltenyi Biotec). Cells were then labeled 30 min at 4 °C with the following antibodies:

for progenitors cells: Biotin-Lin (mix of anti-CD4, CD5, CD8a, Mac-1, B220, Ter119, Gr-1 antibodies, all from BD Bioscience); Kit APC (Biolegend # 105812); Sca 1 BV421 (Biolegend #108128); CD41 APC-Cy7 (Biolegend #133928); CD34 FITC (Biolegend #553733); CD16/32 BV510 (Biolegend #101333) for precursors and mature cells: CD41 APC-Cy7 (Biolegend #133928); Ter119 PE-Cy7 (Biolegend #116222); Gr1 FITC (Biolegend #108406); CD3 PE (Biolegend #100206); CD19 PB (Biolegend #115523). After washing, Steptavidine PerCP-Cy5.5 was added during 15 minutes at 4 °C and cells were then sorted on a FACS Aria LSR (Becton Dickcinson).

The following populations were isolated: LSK (Lin- Kit+ Sca + ); Megakaryocytes-Erythroid Progenitors (MEP) (Lin- Kit+ Sca- CD CD34-CD16/32-41-); Granulocyte-Monocyte Progenitors (GMP): Lin- Kit+ Sca-CD41- CD34 + CD16/32 + ; Megakaryocytic progenitors (MkP, Lin- Kit+ Sca- CD41 + ); T Lymphocytes (CD3 + ); B Lymphocytes (CD19 + ); Granulocytic precursors (CD19-CD3-Gr1 + ); Erythroid precursors (CD19-CD3-Ter119 + ).

Cells were frozen at −80 °C and DNA extracted using Nucleospin Tissue extraction kit (Macherey Nagel). The JAK2V617F allele burden was determined using a droplet digital PCR technic (Biorad QX200 system). The following primer and probes were used:

mJAK2-FwdCGAAGCAGCAAGCATGATGAG
mJAK2-RevGAGAGTAAGTAAAGCCAcCTGCT
mJAK2-Probe WTHEX-ATTATGGTGTCTGTGTCTGTGGAG-BHQ
Fwd mJAK2 ddPCRAAGCCCAGGTGATGGATGAGG
Rev mJAK2 ddPCRGTTACACGAGTCACCCATAATC
Probe mJAK2V617F ddPCRFAM-TACGAAGTTATTGGATTTCCTGCG-BHQ

The amplification program was as follows: 95 °C during 10′, before 40 cycles at 94 °C during 30′ and 57 °C during 1′, and a final step at 98 °C during 10′. Results were analyzed with Quantasoft software and expressed as the proportion of mutated allele among the number of copies of total JAK2 detected.

### Bulk RNA sequencing & analysis

After RNA extraction, RNA concentrations were obtained using nanodrop or a fluorometric Qubit RNA assay (Life Technologies, Grand Island, New York, USA). The quality of the RNA (RNA integrity number) was determined on the Agilent 2100 Bioanalyzer (Agilent Technologies, Palo Alto, CA, USA) as per the manufacturer's instructions.

To construct the libraries, 100 ng of high quality total RNA sample (RIN > 8) was processed using NEBNext Ultra II RNA Library Prep Ki (New England BioLabs) according to manufacturer instructions. Briefly, after purification of poly-A containing mRNA molecules, mRNA molecules are fragmented and reverse- transcribed using random primers. Replacement of dTTP by dUTP during the second strand synthesis will permit to achieve the strand specificity. Addition of a single A base to the cDNA is followed by ligation of Illumina adapters. Libraries were quantified by qPCR using the KAPA Library Quantification Kit for Illumina Libraries (KapaBiosystems, Wilmington, MA) and library profiles were assessed using the DNA High Sensitivity LabChip kit on an Agilent Bioanalyzer. Libraries were sequenced on an Illumina Nextseq 500 instrument using 75 base-lengths read V2 chemistry in a paired-end mode. After sequencing, a primary analysis based on AOZAN software (ENS, Paris) was applied to demultiplex and control the quality of the raw data (based of FastQC modules/version 0.11.5). Obtained fastq files were then aligned using STAR algorithm (version 2.5.2b) and quality control of the alignment realized with Picard tools (version 2.8.1). Reads were then counted using Featurecount (version Rsubread 1.24.1) and the statistical analyses on the read counts were performed with the DESeq2 package version 1.14.1 to determine the proportion of differentially expressed genes between two conditions.

Raw fastq files were mapped to mouse genome GRCm38 release 96 with STAR 2.7.1a (Spliced Transcripts Alignment To a ref. [48]. Normalization and differential expression analysis was performed with DESeq2 1.32.0 package (https://doi.org/10.1186/s13059-014-0550-8). Adjusted p-values were calculated using Benjamini and Hochberg's approach, and genes with adjusted p-value <0.05 were considered as differentially expressed and selected for functional enrichment using clusterProfiler package v4.0.5 (https://doi.org/10.1093/bioinformatics/btq064).

### CITE-seq

Jak2^WT (n = 5) and Jak2^V617F Myel (n = 5) mice under isoflurane anesthesia received 2 μg of anti-CD45.2-APC (clone 104, BioLegend) i.v. 5 min before sacrifice to label circulating immune cells[19] and were killed by cervical dislocation. Upon collection of the aorta using a stereomicroscope, aneurysm development was macroscopically detected in 2 Jak2^V617F Myel mice. Aortas were collected and enzymatically digested in RPMI medium containing 450 U/ml collagenase I (Sigma-Aldrich C0130), 125 U/ml collagenase XI (Sigma-Aldrich C7657), 60 U/ml Hyaluronidase (Sigma-Aldrich H3506) and 60U/ml DNAse1 (Roche #11284932001) for 45 min at 37 °C with agitation, washed in PBS + 1% FCS and passed through a 70 μm cell strainer. After washing in PBS + 1% FCS, aortic cells were resuspended in erythrocyte lysis buffer (ACK buffer) for 5 min and washed again with PBS + 1% FCS. Cells were plated in round bottom 96 well plates, and incubated for 10 min on ice with anti-CD16/32 (BioLegend, Clone 93, 10 μg/ml in PBS + 1% FCS) to block unspecific binding of antibodies to Fc receptors. Cells were subsequently labeled in a final volume of 60 μl for 25 minutes at 4 °C with a mix containing CITE-seq[20] antibodies (TotalSeq-A, BioLegend, see table below), 1:1000 Fixable Viability Dye e780 (ThermoFisher Scientific 65-0865-14), 2 μg/ml anti-CD45.2 BV421 (Jak2^WT samples) or 2 μg/ml anti-CD45.2 Alexa488 (Jak2^V617F samples). Each unique sample was

labeled with a specific Hashtag antibody for cell hashing[49] (Hashtag 1 to 5: *Jak2^WT* aortas, Hashtag 7 and 9: *Jak2^V617F Myel* with aneurysm; Hashtag 6, 8, and 10: *Jak2^V617F Myel* without aneurysm; final concentration 1:100, BioLegend TotalSeq-A).

| TotalSeq-A antibody against: | BioLegend Cat# | Dilution |
|---|---|---|
| Ly6G | 127655 | 500 |
| CD11b | 101265 | 500 |
| CD62L | 104451 | 500 |
| IA_IE | 107653 | 500 |
| CD54 (ICAM1) | 116127 | 500 |
| Ly6C | 128047 | 800 |
| CD115 | 135533 | 500 |
| CXCR4 | 146520 | 500 |
| Msr1 | 154703 | 500 |
| CD64 | 139325 | 500 |
| FCeRIa | 134333 | 500 |
| CCR3 | 144523 | 500 |
| CD49d | 103623 | 500 |
| CD80 | 104745 | 500 |
| CD117 | 105843 | 500 |
| Sca1 | 108147 | 500 |
| CD11c | 117355 | 500 |
| TIM4 | 130011 | 500 |
| CX3CR1 | 149041 | 500 |
| XCR1 | 148227 | 500 |
| F4/80 | 123153 | 500 |
| CD86 | 105047 | 500 |
| CD135 | 135316 | 500 |
| CD103 | 121437 | 500 |
| CD169 | 142425 | 500 |
| CD8a | 100773 | 500 |
| SiglecH | 129615 | 500 |
| CD19 | 115559 | 500 |
| CD3 | 100251 | 500 |
| CD63 | 143915 | 500 |
| CD9 | 124819 | 500 |
| CD163 | 155303 | 500 |
| NK1.1 | 108755 | 500 |
| CD127 | 135045 | 500 |
| CD68 | 137031 | 500 |
| Sirpa | 144033 | 500 |
| ITGB7 | 321227 | 500 |
| CD4 | 100569 | 500 |
| CD26 | 137811 | 500 |
| MGL2 | 146817 | 500 |
| TCRgd | 118137 | 500 |
| CCR2 | 150625 | 500 |
| CD44 | 103045 | 500 |
| CD21/35 | 123427 | 500 |
| CD43 | 143211 | 500 |
| Armenian Hamster IgG Isotype Ctrl | 400973 | 500 |
| Rat IgG1, κ Isotype Ctrl | 400459 | 500 |
| Rat IgG2a, κ Isotype Ctrl | 400571 | 500 |
| Rat IgG2b, κ Isotype Ctrl | 400673 | 500 |
| CD47 | 127535 | 500 |
| SiglecF | 155513 | 500 |
| CD137 | 106111 | 500 |
| CD36 | 102621 | 500 |
| CCR5 | 107019 | 500 |
| CD278 | 117409 | 500 |
| PIR-A_B | 144105 | 500 |
| CD5 | 100637 | 500 |
| CD304 | 145215 | 500 |
| CD40 | 124633 | 500 |
| CD14 | 123333 | 500 |
| CD95 | 152614 | 500 |
| CD300c_d | 148005 | 500 |
| IL1RL1 | 145317 | 500 |
| TCRbeta | 109247 | 500 |
| Mac2 | 125421 | 500 |
| CD137L | 107109 | 500 |
| CD178 | 106613 | 500 |
| CD55 | 131809 | 500 |
| CD226 | 128823 | 500 |
| TIGIT | 142115 | 500 |
| CD39 | 143813 | 500 |
| JAML | 128507 | 500 |
| CXCR5 | 145535 | 500 |
| MGL1 | 145611 | 500 |
| CD24 | 101841 | 500 |

After labeling, cells were washed twice in PBS + 1% FCS and pooled. Viable leukocytes were sorted with exclusion of contaminating circulating leukocytes (CD45.2-APC⁺) using a BD FACS Aria III with a 100 μm nozzle. The amount of sorted cells was adjusted so that cells from *Jak2^WT* (CD45.2-BV421⁺) and from *Jak2^V617F* aortas (CD45.2-Alexa488⁺) each represented ~50% of the total sorted cells. Cells were sorted in PBS supplemented with 1% PBS, and washed twice post-sort in PBS + 0.04% UltraPure BSA (ThermoFisher AM2616). Cells were counted in Trypan blue to assess viability (>80%). A total of 20,000 cells were loaded at a concentration of 500 cells/μl in the 10× Genomics Chromium (Single Cell 3′ v3 reagents, 10× Genomics) and scRNA-seq, ADT and HTO libraries prepared and sequenced as described in[26].

### Single-cell RNA-seq data analysis

A total of 10× Genomics data, including HTO and ADT libraries, was demultiplexed using Cell Ranger software (Pipeline Version 6.0.2). Mouse mm10 reference genome was used for the alignment and counting steps. The Feature-Barcode matrix obtained from Cell Ranger was further analyzed in Seurat v3.1.1[50]. The "HTODemux" function was used to identify sample of origin of single cells, exclude multiplets (i.e. cells positive for more than one hashtag signal) and cells with no detectable hashtag signal. Cells with more than 10% mitochondrial transcripts were excluded from further analysis. Data were normalized using the "NormalizeData" function, and 2000 variable features identified with the "FindVariableFeatures" function using the "vst" selection method. Data were scaled using the "Scale-Data" function and principal component (PC) analysis performed. Clustering was performed using the "FindNeighbors" and "FindClusters" functions with 20 PCs and a resolution parameter of 0.3. Uniform Manifold Approximation and Projection (UMAP) dimensional reduction was performed using 20 PCs. Immune cell populations were identified based on CITE-seq signal for established cell surface markers (see Results). Cells corresponding to monocytes/macrophages were subset in a new Seurat object and reclustered. This refined analysis revealed clusters of monocyte/

macrophages with very low RNA content that were considered as low quality/dead cells and removed from further analysis. The curated monocyte/macrophage dataset was analyzed with clustering (resolution 0.8) and dimensional reduction as described above. 2 clusters of $Ccr2^+$ macrophages with a clearly overlapping pro-inflammatory gene expression profile were manually pooled into the "$Ccr2^+$ Inflammatory" macrophage cluster. Enriched transcripts and cell surface markers in monocyte/macrophage populations were determined using the "FindAllMarkers" function. Data visualizations were generated using built-in functions in Seurat (e.g. "FeaturePlot", "DotPlot") and dittoSeq (https://github.com/dtm2451/dittoSeq).

## Quantitative real-time polymerase chain reaction

Thoracic aortas were harvested after flushing with cold PBS and were frozen at −80 °C. Total RNA was extracted from frozen mouse thoracic aorta samples with Qiagen columns (RNeasy MiniSpin Columns) using a polytron (T25 basic, IKA, Labortechnik). The phase containing RNAs was then recuperate and washed with molecular biology water. RNA quality control and concentration were performed using Nanodrop 2000 (Thermofisher scientific). and then stored at −80 °C. Reverse transcription was done following manufacturer instruction (kit QuantiTect Reverse Transcription (Qiagen)).

Real-time fluorescence monitoring was performed with the Applied Biosystems, Step One Plus Real-Time PCR System with Power SYBR Green PCR Master Mix (Eurogentec). qPCR was performed in triplicate for each sample. GAPDH cycle threshold was used to normalize gene expression: (F: 50 -CGTCCCGTAGACAAAATGGTGAA-30; R: 50 - GCC GTGAGTGGAGTCATACTGGAACA-30). Relative expression was calculated using the 2-delta-delta CT method followed by geometric average, as recommended.

The following primer sequences were used:

Il1β: Forward 5′-GAAGAGCCCATCCTCTGTGA-3′; Reverse 5′-GG GTGTGCCGTCTTTCATTA-3′; Tnfα: Forward 5′-GATGGGGGGCTTC CAGAACT-3′, Reverse: 5′-CGTGGGCTACAGGCTTGTCAC-3′, Il6: Forward 5′-TGACAACCACGGCCTTCCCTA-3′, Reverse 5′-TCAGAATTGCC ATTGCACAACTCTT-3′, Il10: Forward 5′-CTCCTAGAGCTGCGGAC TGCCTTCA-3′, Reverse 5′-CTGGGGCATCACTTCTACCAGGTAAAA-3′, Ccl2: forward 5′-CCCCACTCACCTGCTGCTA-3′, reverse 5′-TACGGGTC AACTTCACATTCAAA-3′

## Blood pressure

Non-invasive systolic and diastolic blood pressure monitoring using tail cuff was done using Visitech BP-2000. Mice were trained for blood pressure 2 days prior to taking readings. Blood pressures were measured at 7-week-old. In each animal, the system automatically performed 4 measurements first, which were not recorded, then 10 consecutive measurements that were recorded.

## Ultrasound study

We used Vevo 2100 ultrasound imager (VisualSonics) equipped with MS550D 22–55 MHz transducer. During imaging, each mouse was anesthetized with 2% isoflurane administered through an adapted nose cone. The mouse was then placed in a supine position on a heated platform to maintain its body temperature at approximately 37 °C. The hair over the areas being imaged was removed with a depilatory cream. Degassed ultrasound gel was used as a coupling medium. The ultrasound device was placed on the thorax and then on the abdomen of the mouse. Sagittal and transversal images were acquired.

## Statistical analysis

Values are expressed as mean ± SEM. Differences between values were examined using the nonparametric two-tailed Mann-Whitney, Kruskal-Wallis tests when appropriate and were considered significant at $P < 0.05$.

## Reporting summary

Further information on research design is available in the Nature Portfolio Reporting Summary linked to this article.

## Data availability

CITE-seq and single-cell RNA-sequencing data generated for this report has been deposited in Gene Expression Omnibus (https://www.ncbi.nlm.nih.gov/geo/query/acc.cgi?acc=GSE214880). Bulk RNA sequencing data generated for this report has been deposited in Gene Expression Omnibus (https://www.ncbi.nlm.nih.gov/geo/query/acc.cgi?acc=GSE215455).

All other data generated or analysed during this study are included in this published article (and its supplementary information files). Source data are provided with this paper.

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

## Acknowledgements

This work was supported by Inserm (Z.M., S.T., A.T., H.A.O.), Fondation Lefoulon-Delalande (RAR) and the British Heart Foundation (Z.M.). We would like to thank Panagiota Arampatzi and the Core Unit Systems Medicine of the University Würzburg for assistance with single-cell RNA-seq analysis.

## Author contributions

R.A.R. and M.V. conducted experiments, acquired data, analyzed data, performed statistical analysis and wrote the manuscript. J.R.L., T.M., J.C., J.P., L.L., B.E., C.K., J.V. conducted experiments, acquired data, analyzed data. P.H., F.F., L.A. and M.L. participate in human study. O.M. and C.J. provided transgenic mice. M.D. analysis transcriptomic profile in the aorta of Jak2V617F HC-EC mice. A.Z., G.R., A.E.S., and C.C. performed scRNA seq experiments and analysis. A.T., Z.M., S.T., P.E.R. drafted of the manuscript. H.A.O. designed the experiments, analyzed data, performed statistical analysis, drafted of the manuscript and obtained funding for the project. M.D. and R.B. performed statistical analysis, All authors discussed and critically revised the manuscript.

## Competing interests

The authors declare no competing interests

## Additional information

[1]Université Paris Cité, Inserm, PARCC, F-75015, Paris, France. [2]Service de médecine vasculaire, Hopital Européen G. Pompidou, Paris, France. [3]Service de gériatrie, Hopital Européen G. Pompidou, Paris, France. [4]Centre de recherche sur l'inflammation, Inserm, Paris, France. [5]Université de Bordeaux, UMR1034, Inserm, Biology of Cardiovascular Diseases, CHU de Bordeaux, Laboratoire d'Hématologie, Pessac, France. [6]Laboratoire d'Hématologie, Hôpital Saint-Antoine, AP-HP, Paris, France. [7]Institute of Experimental Biomedicine, University Hospital Wuerzburg, Würzburg, Germany. [8]Helmholtz Institute for RNA-based Infection Research (HIRI), Helmholtz-Center for Infection Research (HZI), Würzburg, Germany. [9]Service de radiologie, Hôpital Saint-Antoine, AP-HP, Paris, France. [10]GlandOmics, 41700 Cheverny, & Department of Diabetology, AP-HP, Hôpital Cochin, Paris, France. [11]AP-HP, Hôpital Beaujon, Service d'Hé-patologie, DMU DIGEST, Centre de Référence des Maladies Vasculaires du Foie, FILFOIE, ERN RARE-LIVER, Clichy, France. [12]Medical Intensive Care Unit, Hôpital Saint-Antoine, AP-HP, Sorbonne Université, Paris, France. [13]These authors contributed equally: Rida Al-Rifai, Marie Vandestienne. ✉e-mail: hafid.aitoufella@inserm.fr

