## [Peer Review File · Nature Communications]

JAK2V617F mutation drives vascular resident macrophages toward a pathogenic phenotype and promotes dissecting aortic aneurysmReviewers' comments:

Reviewer #1 (Remarks to the Author):

In this manuscript, the authors studied the role of JAK2V617F mutation in aortic aneurysm using Cre-mediated expression of JAK2V617F in mouse models. The authors show that hematopoietic and endothelial expression of JAK2V617F promotes infiltration of macrophages in the aorta and induces a pro-inflammatory phenotype resulting in aortic dilation and dissecting aortic aneurysm. Although the authors have carried out a series of experiments with complementary models to address their hypothesis, several prior published studies (PMID: 30571460, 31709318, 33731931 and 33567809) have already suggested a role for JAK2V617F-induced proliferation/infiltration of macrophages and inflammatory cytokine production in cardiovascular diseases including atherosclerosis, aortic lesions and aortic aneurysms. The JAK2V617F mutation has been frequently observed in MPN. JAK2V617F mutation also has been found in healthy individuals with clonal hematopoiesis of indeterminate potential (CHIP) who do not exhibit MPN phenotype but at increased risk of developing cardiovascular abnormalities. Although the authors suggested that JAK2V617F-dependent clonal hematopoiesis promotes dissecting aortic aneurysm, most experiments in this manuscript are conducted in the context of JAK2V617F-induced MPN rather than clonal hematopoiesis.

Major points

1. In Fig. 1A, it is not clear how many JAK2V617F-positive individuals without MPN diagnosis has aortic aneurysm. Since the overall conclusion is that JAK2V617F-dependent clonal hematopoiesis promotes aortic aneurysm, it is important to have a correlation of this disease with JAK2V617F-positive individuals without MPN diagnosis. Otherwise, it will be seen as a complication associated with MPN, which has many other cancer-related complications.
2. The experimental design of this study does not appropriately address whether JAK2V617F-dependent clonal hematopoiesis drives aortic aneurysm. Current experimental setup does not distinguish between clonal hematopoiesis and MPN. It appears from this study that aortic aneurysm is a complication associated with JAK2V617F-induced MPN in mouse models. Authors should perform competitive transplantation using low percentage of JAK2V617F-positive hematopoietic stem/progenitors to study JAK2V617F-induced clonal hematopoiesis and aortic disease development.
3. JAK2V617F mutated patients have the least association with Abdominal aorta dilation whereas JAK2V617F mice exhibit the most significant changes in Abdominal aorta diameter (in Fig. 1A and 1F) indicating that mice studies do not recapitulate the phenotype observed in humans.
4. Fig. 1G shows reduced collagen expression in the aorta of JAK2V617F-expressing mice. Previous studies showed increased Col1a1 and Col3a1 expression in the myocardium of JAK2V617F-expressing mice. The authors should perform qPCR analysis to examine collagen expression in the aorta of these mice in order to get a more quantitative assessment. They should also provide qPCR data on collagen expression in Fig. 3F and Fig. 5F. Similarly, MMP-2, -3, -9 and -13 expression should be checked by qPCR in Fig. 1I, 3G and 5F.
5. Mice expressing JAK2V617F exhibit decreased survival (in Fig. 1, 3 and 5). This is not a new observation. Previous studies involving JAK2V617F-induced MPN also had similar observations. The authors did not confirm whether reduced survival in mice expressing JAK2V617F was mainly due to aortic aneurysm.
6. In Fig. 5, the authors crossed JAK2V617F flex mice with Pf4-Cre mice to selectively induce JAK2V617F expression in vascular tissue-resident macrophages. Pf4 is highly expressed in megakaryocytes/platelets. Hence, Pf4-Cre mouse was originally developed to selectively delete or induce expression in the megakaryocytes/platelets. Pf4 is also expressed in hematopoietic stem/progenitors and other myeloid cells at a lower level. In Fig. 5D, it is clearly seen that Pf4-cre-mediated expression of Jak2V617F resulted in increased platelets. The authors claim that Jak2V617F

Flex; Pf4Cre mice represent resident macrophage-dependent Jak2V617F expression is misleading. If the results from Figure 5 are true, it will suggest that megakaryocytes/platelets expressing Jak2V617F also have important contribution in the development of aortic aneurysm. What is the status of CD41+ megakaryocytic population in the aorta of these mice?

7. In Fig 5K-Q, the authors suggested that EPO treatment increases aortic aneurysm. EPO is important for erythropoiesis. What is the contribution of erythroid cells in aortic aneurysm? Also, it is not clear how EPO could increase aortic aneurysm. Possible mechanistic explanation should be provided.

8. In Fig. 6, the authors tested the efficacy of CSF-1 inhibitor Ki20227 and JAK2 inhibitor ruxolitinib in reducing aortic disease severity in Jak2V617F mice. The JAK2V617F mutant mice without treatment (vehicle) group is missing in Fig. 6G-I. Without these controls, you cannot make any comparison whether CSF-1 inhibitor Ki20227 treatment reduces aortic disease. Does Ki20227 treatment completely prevent deaths in Jak2V617F mice (in Fig. 6F). If so, CSF-1 inhibitor Ki20227 is more effective than JAK2 inhibitor ruxolitinib in preventing aortic disease. Does CSF-1 inhibitor Ki20227 treatment also significantly reduce the expression of inflammatory cytokines. The authors should carefully compare CSF-1 inhibitor Ki20227 with ruxolitinib in the same experiment to suggest which treatment is better for prevention of aortic disease induced by JAK2V617F mutation.

Reviewer #2 (Remarks to the Author):

Al-Rifai et al. reported that perivascular tissue-resident macrophage with Jak2VF mutation polarized toward pro-inflammatory phenotype and promoted deleterious aortic wall remodeling and dissecting aneurysm. They also showed that aortic resident macrophages expressed EPOR, and EPO treatment induced local increase in proliferating resident macrophages, enhanced vascular pro-inflammatory responses, and promoted aortic dilation in WT mice. Ki20227 (CSF1 inhibitor) and ruxolitinib treatment depleted aortic macrophages and alleviated lethal aorta rupture in Jak2VF MyC mice. Mice work was done detailed and precisely, and their results were convincing. Their work shed the light on the vascular remodeling and development aorta dilatation and aneurysm induced by JAK2VF mutation, observed in MPN patients and perhaps individuals with normal blood cell count and CHIP.

Authors demonstrated that the JAK2VF mutation was associated with dilatation of both ascending and descending thoracic aorta. They also showed perivascular tissue-resident macrophages with Jak2VF caused dissecting aneurysm in mice model. In patients' aneurysms, did resident macrophages harbor JAK2VF mutation?

Are enhancing vascular pro-inflammatory responses observed in MacR mice?

Many chronic renal failure and MDS patients received EPO administration. Is there any report EPO induced aneurysms in them?

Authors demonstrated aneurysm development by tissue-resident macrophages with Jak2VF using transgenic mice. Does aorta dilatation or aneurysm development occur when the mixture of BM cells from Jak2VF EC mice and BM cells from WT mice were transplanted (mimicking CHIP)?

Reviewer #3 (Remarks to the Author):

In this study, the authors used several genetic mouse models and a set of ex vivo functional studies to show that JAK2V617F mutation drives adventitial tissue-resident macrophages toward a pathogenic inflammatory phenotype causing aortic aneurysm and dissection. The logic flow is fine,

and the amount of work involved in this study is considerable. It is regrettable that the novelty of the study has been compromised as the role of hematopoietic JAK2V617F mutation and the corresponding activation of macrophages in the pathogenesis of aortic aneurysm and the mitigating effect of Jak2 inhibitor ruxolitinib have been recently reported (Haematologica. 2021 Jul 1;106(7):1910-1922). The novelty of this study is thus to highlight the initiation role of vascular resident macrophages in such aortopathy. The view is interesting, but the evidence still needs to be strengthened. Some comments are as follows:

- 1) The authors used Pf4-Cre to generate transgenic mice with a selective JAK2V617F mutation in the presumed vascular tissue-resident macrophages, which is a very important strategy for gaining new insights. From the view of scRNA-seq data (Figure 5B), expression of Pf4 is not confined to the three tissue-resident macrophage populations. Therefore, in order to draw the conclusion that “the JAK2V617F mutation in vascular tissue-resident macrophages induces dissecting aortic aneurysm”, the specificity of Pf4-Cre-mediated ablation in aortic tissue-resident macrophages but not in other macrophage populations should be validated.
- 2) If the authors propose a pivotal role of JAK2V617F mutation in vascular tissue-resident macrophages in the formation of aortic aneurysm, such aortopathy might be irrelevant to clonal hematopoiesis, since it has been reported that the cell origin of vascular tissue-resident macrophages is not from adult bone marrow hematopoiesis (Nat Immunol. 2016 Feb;17(2):159-68). It should be very careful to say “JAK2V617F-dependent clonal hematopoiesis drives vascular resident macrophages toward a pathogenic phenotype and promotes dissecting aortic aneurysm” in title, unless the authors provide evidence that these vascular tissue-resident macrophages are the progenies of clonal hematopoiesis.
- 3) As the authors mentioned in line 415, a recent study showed that EPO injection promoted aortic aneurysm in Apoe^{-/-} mice and suggested that endothelial cell activation might be involved in EPO-induced abdominal aortic aneurysm. Whether endothelial cell activation also exists in the aortic aneurysm induced by high dose of EPO injection in the present study? If the authors could not determine the specificity of the responding cell types, I suggest this part of results to be removed from the current MS.
- 4) In Figure 5G, representative immunostaining of Lyve1 should be displayed.

Reviewer #4 (Remarks to the Author):

This study tested a role for Jak2 V617F mutant form in residential macrophages in mouse aortic (ascending, descending, and abdominal) aneurysms. The role of this mutant form of Jak2 has been tested in few other cardiovascular disease models. Overall, many results are descriptive. Here are few of my concerns.

Page 6: It is unexplained why authors generated the Jak2-V617F-VE-Cadherin-Cre^{+/-} mice. Don't we think VE-Cadherin is an EC protein? In other word, authors initially might want to study Jak2-V617F function in EC? Please provide the background information of this experiment.

Figure 2: In this Figure, authors performed the BMT using bone marrows from WT mice to irradiated WT and Jak2-V617F-HC-EC mice and claimed no role of non-myeloid cells in aortic aneurysm development. Since data are all negative, it is not necessary to show these data in the main text. Authors should move the whole figure into the supplementary file.

Data in Figure 4 did not give much useful information either, besides authors used the CITE-seq technique. It is common sense that residential macrophages present in the aortas. These data can also be in the supplementary file. Pf4 is a marker of residential macrophages. This is known. Straight use of Pf4-Cre mice will make much more sense.

Uses of the LysM-Cre^{+/-} mice and Pf4-Cre mice to study macrophages are valid and sufficient. However, use EPO as an activator of vascular residential macrophages is not a clean experiment. EPO receptor is expressed not only in macrophages, but also in many other cell types (PMID: 23874302, PMID: 26349059, PMID: 21357707). The results from Figure 5L to 5O were not necessary all come from macrophage activation.

Results from Ki20227 in Figure 6B to 6I are also questionable, because Ki20227 not only affects residential macrophages, but also many others mechanisms (PMID: 19398755, PMID: 18085662, PMID: 22186992).

Minor comments:

Figure 1F: why the aneurysm development differed between ascending aortas and descending aortas? Any explanation?

It is strange that the authors quantified elastin contents by the numbers of elastin layers. It makes more sense by quantify their contents by immunostaining, or count the numbers of elastin breaks.

Reviewer #1 (Remarks to the Author):

In this manuscript, the authors studied the role of JAK2V617F mutation in aortic aneurysm using Cre-mediated expression of JAK2V617F in mouse models. The authors show that hematopoietic and endothelial expression of JAK2V617F promotes infiltration of macrophages in the aorta and induces a pro-inflammatory phenotype resulting in aortic dilation and dissecting aortic aneurysm. Although the authors have carried out a series of experiments with complementary models to address their hypothesis, several prior published studies (PMID: 30571460, 31709318, 33731931 and 33567809) have already suggested a role for JAK2V617F-induced proliferation/infiltration of macrophages and inflammatory cytokine production in cardiovascular diseases including atherosclerosis, aortic lesions and aortic aneurysms. The JAK2V617F mutation has been frequently observed in MPN. JAK2V617F mutation also has been found in healthy individuals with clonal hematopoiesis of indeterminate potential (CHIP) who do not exhibit MPN phenotype but at increased risk of developing cardiovascular abnormalities. Although the authors suggested that JAK2V617F-dependent clonal hematopoiesis promotes dissecting aortic aneurysm, most experiments in this manuscript are conducted in the context of JAK2V617F-induced MPN rather than clonal hematopoiesis.

Response: We would like to thank the reviewer for his/her comments that clearly helped improve the manuscript. Indeed, we showed that JAK2V617F mutation promotes dissecting aneurysm in murine models with myeloproliferative neoplasm (MPN). Myeloproliferative neoplasm is a prototypic clonal hematopoiesis-related disease, but to avoid any misunderstanding with clonal hematopoiesis of indeterminate potential (CHIP) that is also associated with the JAK2V617F mutation and to address the reviewer's concern, reference to "clonal hematopoiesis" has been removed in the manuscript and we clarified that our study reports severe aortopathy in the context of JAK2+myeloproliferative neoplasm.

Here using a step-by-step approach with several complementary approaches we showed that JAK2V617F mutation affects vascular resident macrophages and drives their phenotype toward a pathogenic profile, promoting both aortic aneurysm formation and complications.

Major points

1. In Fig. 1A, it is not clear how many JAK2V617F-positive individuals without MPN diagnosis has aortic aneurysm. Since the overall conclusion is that JAK2V617F-dependent clonal hematopoiesis promotes aortic aneurysm, it is important to have a correlation of this disease with JAK2V617F-positive individuals without MPN diagnosis. Otherwise, it will be seen as a complication associated with MPN, which has many other cancer-related complications.

Response : In our study, we compared JAK2V617F+ patients with MPN and age- and gender-matched controls without JAK2 mutation. Unfortunately, we did not have access to JAK2V617F+ patients without MPN diagnosis.

To address the reviewer's concerns, "clonal hematopoiesis" has been removed and we clarified that we only described vascular phenotype of JAK2V617F+ patients with MPN.

2. The experimental design of this study does not appropriately address whether JAK2V617F-dependent clonal hematopoiesis drives aortic aneurysm. Current experimental setup does not distinguish between clonal hematopoiesis and MPN. It appears from this study that aortic aneurysm is a complication associated with JAK2V617F-induced MPN in mouse models. Authors should perform competitive transplantation using low percentage of JAK2V617F-positive hematopoietic stem/progenitors to study JAK2V617F-induced clonal hematopoiesis and aortic disease development.

Response: We do apologize for this confusion. We did not aim to evaluate clonal hematopoiesis in the absence of myeloproliferative neoplasm. Yet, as suggested to address the reviewer's concern, we performed an additional series of experiments. Irradiated C57Bl6 mice were transplanted with 100% Jak^{wt} BM cells or a mixture of 80% Jak^{wt}/20% Jak2^{V617F} HC-EC BM cells. One month after BMT, we observed significant aorta dilation in chimeric mice receiving 20% Jak2^{V617F} BM cells, despite no major changes in blood cell subsets (Supplementary Figure 7B-C).

3. JAK2V617F mutated patients have the least association with Abdominal aorta dilation whereas JAK2V617F mice exhibit the most significant changes in Abdominal aorta diameter (in Fig. 1A and 1F) indicating that mice studies do not recapitulate the phenotype observed in humans.

Response : In our work, we used several complementary models to decipher how JAK2V617F mutation induced spontaneous dissecting aortopathy. In patients, we found that JAK2V617F mutation was associated with significant changes in aortic diameter of the ascending and descending thoracic aorta, but not abdominal aorta (even though a trend was observed). Regarding comparison with genetically-modified mouse models, we found:

-In HC-EC model, an increase in descending aorta

-In chimeric HC-EC model, an increase in ascending, descending aorta, as well as in the abdominal region.

-In Myc model, an increase in ascending, descending aorta, as well as in the abdominal region.

-In MacR model, an increase in ascending, descending aorta, as well as in the abdominal region.

Altogether, we can conclude that JAK2V617F mutation promoted aortopathy affecting the ascending and descending aorta in both mice and human.

As detailed in the discussion section, the discrepancy of severity between human and murine aortic disease might be accounted for by the fact that JAK2V617 mutation in human is a somatic mutation occurring at late stages in adult, whereas it is a germline mutation in mice. Another important argument for similarities between human and murine disease is that, as shown in figure 6J-O, drugs currently used clinically to treat JAK2V617+ myeloproliferative diseases such as ruxolitinib, significantly limited thoracic aorta dilation and decreased lethal aortic aneurysm rupture in JAK2V617 mutant mice.

4. Fig. 1G shows reduced collagen expression in the aorta of JAK2V617F-expressing mice. Previous studies showed increased Col1a1 and Col3a1 expression in the myocardium of JAK2V617F-expressing mice. The authors should perform qPCR analysis to examine collagen expression in the aorta of these mice in order to get a more quantitative assessment. They should also provide qPCR data on collagen expression in Fig. 3F and Fig. 5F. Similarly, MMP-2, -3, -9 and -13 expression should be checked by qPCR in Fig. 1I, 3G and 5F.

Response: In the study which the Reviewer mentioned, Sano et al. reported increased expression of collagen genes in the heart in the context of myocardial infarction and transverse aortic constriction (JACC Basic 2019) which is very different than our models. As requested, the quantification of collagen and MMP mRNA levels by qPCR has been done and added in each figures (Figures 2G&I, 5H&I) as well as in supplemental figures (Suppl figures 2D&F, 12I&M). Overall, we found no change in Cola1 and Cola3 gene expression between control and mutant mice, but MMP gene and activity were increased. These results support that decreased collagen content observed in the aortic adventitia of mutant mice was due to increased degradation by MMPs and not to reduced production.

5. Mice expressing JAK2V617F exhibit decreased survival (in Fig. 1, 3 and 5). This is not a new observation. Previous studies involving JAK2V617F-induced MPN also had similar observations. The authors did not confirm whether reduced survival in mice expressing JAK2V617F was mainly due to aortic aneurysm.

Response: We do agree with the reviewer that increased mortality in JAK2 mutant mice has been previously reported, which is consistent with our observations. However, the causes of mortality remained totally unknown. For example, in the study by Kubovcakova (Blood 2013) the authors wrote: "The cause of death in most of the mice remains unclear. The mice did not appear to be visibly sick before they died and there were no signs of leukemic transformation." As detailed in our manuscript, survival was monitored and necropsy was systematically performed in dead mice. We found in almost all analyzed animals, blood in the thorax and/or in the abdominal cavity due to dissecting aortic aneurysm. Moreover, all interventions (genetic or pharmacological) that showed protection against aortic dilation in JAK2 mutant mice also increased survival.

6. In Fig. 5, the authors crossed JAK2V617F flex mice with Pf4-Cre mice to selectively induce JAK2V617F expression in vascular tissue-resident macrophages. Pf4 is highly expressed in megakaryocytes/platelets. Hence, Pf4-Cre mouse was originally developed to selectively delete or induce expression in the megakaryocytes/platelets. Pf4 is also expressed in hematopoietic stem/progenitors and other myeloid cells at a lower level. In Fig. 5D, it is clearly seen that Pf4-cre-mediated expression of Jak2V617F resulted in increased platelets. The authors claim that Jak2V617F Flex; Pf4Cre mice represent resident macrophage-dependent Jak2V617F expression is misleading. If the results from Figure 5 are true, it will suggest that megakaryocytes/platelets expressing Jak2V617F also have important contribution in the development of aortic aneurysm. What is the status of CD41+ megakaryocytic population in the aorta of these mice?

Response: We would like to thank the reviewer for the detailed analysis of our work. We used several complementary murine models to decipher step-by-step the mechanisms responsible for dissecting aneurysm. LysMCre Jak2 Lox/lox mice (named Myc model) had normal platelet counts in blood but developed severe aortopathy responsible for 100% mortality at 16 weeks of age (Figure 2). These results clearly rule out the possibility that platelets contribute in the pathophysiology of JAK2V617F-induced aortopathy in mutant mice.

Using Sc RNA seq, we did not detect megakaryocytes in the aorta of mutant mice. On peut aussi citer le modèle EPO

7. In Fig 5K-Q, the authors suggested that EPO treatment increases aortic aneurysm. EPO is important for erythropoiesis. What is the contribution of erythroid cells in aortic aneurysm? Also, it is not clear how EPO could increase aortic aneurysm. Possible mechanistic explanation should be provided.

Response: We would like to thank the reviewer for this comment. As depicted in figure 5, Pf4Cre Jak2lox/lox developed severe aortopathy despite normal red blood cell counts, which highly suggest that RBC were not involved in JAK2-related aortopathy. In addition, in HC-EC Jak2 mice, we found that the correction of polycythemia with phenylhydrazine (Zhao JCI 2018) had no impact on aorta dilation and rupture. These new data have been added as a new supplementary fig 3.

Given that vascular resident macrophages highly and specifically express EPOR (Figure 5 & supplemental fig 11), we assumed that EPO should activate vascular resident macrophages and promote a pathogenic pro-inflammatory phenotype. Indeed, as depicted in supplementary

figure 12, EPO treatment induced macrophage proliferation in the adventitia and increased pro-inflammatory cytokine and MMP production in the aortic wall.

8. In Fig. 6, the authors tested the efficacy of CSF-1 inhibitor Ki20227 and JAK2 inhibitor ruxolitinib in reducing aortic disease severity in Jak2V617F mice. The JAK2V617F mutant mice without treatment (vehicle) group is missing in Fig. 6G-I. Without these controls, you cannot make any comparison whether CSF-1 inhibitor Ki20227 treatment reduces aortic disease.

Response: As depicted in figure 6, 3 groups of animals were included in the experiment: Jak2wt control mice, non-treated Jak2 mutant and Ki20227-treated Jak2 mutant mice. All non-treated Jak2 mutant mice died because of aorta dissection and could not be included in histological analysis at the end of the experiment (15-week timepoint). Elastin layers and mean aorta diameter were quantified in survivors and we did not find any difference between Jak2wt control mice and Ki20227-treated Jak2 mutant mice.

Does Ki20227 treatment completely prevent deaths in Jak2V617F mice (in Fig. 6F).

Response: As shown in Figure 6, Ki20227 treatment fully prevented mutant mice death.

If so, CSF-1 inhibitor Ki20227 is more effective than JAK2 inhibitor ruxolitinib in preventing aortic disease. Does CSF-1 inhibitor Ki20227 treatment also significantly reduce the expression of inflammatory cytokines. The authors should carefully compare CSF-1 inhibitor Ki20227 with ruxolitinib in the same experiment to suggest which treatment is better for prevention of aortic disease induced by JAK2V617F mutation.

Response: Indeed, Ki20227 treatment abolished the pro-inflammatory signature in the aortic wall of mutant mice. These new results were added as supplementary figure 6H.

Jak2 Myc mice were treated with Ki20227 (figure 6 E-F) or Ruxolitinib (Figure 6M-N). We found that Ki20227 fully prevented aorta rupture and death, whereas Ruxolitinib only reduced death incidence. Based on both experiments, we can conclude that depletion of vascular resident macrophages using Ki20227 was more protective than blocking JAK2 with Ruxolitinib. Please note that Ruxolitinib is currently used clinically in JAK2 patients with MPN, whereas Ki20227 is not approved to treat patients.

Reviewer #2 (Remarks to the Author):

Al-Rifai et al. reported that perivascular tissue-resident macrophage with Jak2VF mutation polarized toward pro-inflammatory phenotype and promoted deleterious aortic wall remodeling and dissecting aneurysm. They also showed that aortic resident macrophages expressed EPOR, and EPO treatment induced local increase in proliferating resident macrophages, enhanced vascular pro-inflammatory responses, and promoted aortic dilation in WT mice. Ki20227 (CSF1 inhibitor) and ruxolitinib treatment depleted aortic macrophages and alleviated lethal aorta rupture in Jak2VF MyC mice. Mice work was done detailed and precisely, and their results were convincing. Their work shed the light on the vascular remodeling and development aorta dilatation and aneurysm induced by JAK2VF mutation, observed in MPN patients and perhaps individuals with normal blood cell count and CHIP. Authors demonstrated that the JAK2VF mutation was associated with dilatation of both ascending and descending thoracic aorta. They also showed perivascular tissue-resident macrophages with Jak2VF caused dissecting aneurysm in mice model.

Response: we would like to thank the reviewer for his positive and encouraging comments.

In patients' aneurysms, did resident macrophages harbor JAK2VF mutation?

Response: We agree with the Reviewer that this is an important point. Unfortunately, we cannot have access to aortic tissue from patients with JAK2 mutation.

Are enhancing vascular pro-inflammatory responses observed in MacR mice?

Response: qPCR analysis of aorta confirmed a vascular pro-inflammatory signature in MacR mice, associated with higher expression of IL-6, TNFa, and MMP transcripts. These new data have been added in figure 5K

Many chronic renal failure and MDS patients received EPO administration. Is there any report EPO induced aneurysms in them?

Response: We would like to thank the reviewer for this valuable comment. Indeed, some trials have reported that chronic renal failure is a risk factor for aortic aneurysm (Matsushita, atherosclerosis 2018) and the association remains significant after adjustment on classical AA risk factors. However, the implication of EPO supplementation, a common treatment for patients with chronic renal failure, in the increased vascular risk for CKD patients cannot be affirmed.

Authors demonstrated aneurysm development by tissue-resident macrophages with Jak2VF using transgenic mice. Does aorta dilatation or aneurysm development occur when the mixture of BM cells from Jak2VF EC mice and BM cells from WT mice were transplanted (mimicking CHIP)?

*Response: To address the reviewer's concern, we performed an additional series of experiments. Irradiated C57Bl6 mice were transplanted with 100% Jak^{wt} BM cells or a mixture of 80% Jak^{wt}/20% Jak2^{V617F} HC-EC BM cells. One month after BMT, we observed a significant aorta dilation in chimeric mice receiving 20% Jak2^{V617F} BM cells, despite no major changes in blood cell subsets (**Supplementary Figure 7B-C**).*

Reviewer #3 (Remarks to the Author):

In this study, the authors used several genetic mouse models and a set of ex vivo functional studies to show that JAK2V617F mutation drives adventitial tissue-resident macrophages toward a pathogenic inflammatory phenotype causing aortic aneurysm and dissection. The logic flow is fine, and the amount of work involved in this study is considerable. It is regrettable that the novelty of the study has been compromised as the role of hematopoietic JAK2V617F mutation and the corresponding activation of macrophages in the pathogenesis of aortic aneurysm and the mitigating effect of Jak2 inhibitor ruxolitinib have been recently reported (Haematologica. 2021 Jul 1;106(7):1910-1922). The novelty of this study is thus to highlight the initiation role of vascular resident macrophages in such aortopathy. The view is interesting, but the evidence still needs to be strengthened.

Response: We would like to thank the reviewer for his/her positive and encouraging comments. We do agree with the reviewer that a recent study suggested a link between JAK2V617F mutation and Aortic aneurysm (Haematologica. 2021 Jul 1;106(7):1910-1922). However, the authors used JAK2V617F bone marrow cell transplantation in Apoe KO mice infused with angiotensin II, which is very far from the clinical setting. It is noteworthy that in our study mice expressing the human JAK2V617F mutation spontaneously developed dissecting aortic aneurysm in the absence of any angiotensin II challenge.

Some comments are as follows:

1) The authors used Pf4-Cre to generate transgenic mice with a selective JAK2V617F mutation in the presumed vascular tissue-resident macrophages, which is a very important strategy for gaining new insights. From the view of scRNA-seq data (Figure 5B), expression of Pf4 is not confined to the three tissue-resident macrophage populations. Therefore, in order to draw the conclusion that “the JAK2V617F mutation in vascular tissue-resident macrophages induces dissecting aortic aneurysm”, the specificity of Pf4-Cre-mediated ablation in aortic tissue-resident macrophages but not in other macrophage populations should be validated.

Response: We agree that expression of Pf4 is not entirely restricted to bona fide tissue resident macrophages but is also found (albeit at a lower level) in other tissue infiltrating macrophages. We have reworded the text to indicate that this model represents rather a strategy to induce ‘preferential recombination in tissue macrophages’. However, the main reason for using this approach is that Pf4 is not expressed in circulating monocytes or their progenitors. We did not detect Pf4 in circulating monocytes by scRNA-seq as shown in supplementary Figure 11. Interrogation of bulk RNA-seq data from ImmGen also indicated that Pf4 is not expressed in circulating monocytes but is highly expressed in tissue resident macrophage populations from the peritoneum and adipose tissue (Reviewer Figure 1A). Pf4 is also not detected in monocyte progenitors in the bone marrow (see below, Reviewer Figure 1B). The preferential recombination pattern in tissue resident macrophages vs. circulating monocytes in Pf4-cre mice has been characterized in the literature (see McKinsey, 2020; Abram, 2014). Therefore, together with our other strategies to induce the JAK2V617F mutation in myeloid cells (Myc Model), the Pf4-cre model provides strong evidence that the observed aortopathy phenotype is promoted by tissue macrophages and not by ‘classical’ monocytes and their progenitors.

Reviewer Figure 1: expression of Pf4 in monocytes, monocyte progenitors and tissue resident macrophages. A) ImmGen Data (<http://rstats.immgen.org/Skyline/skyline.html>) showing expression of Pf4 measured by RNA-seq in the indicated populations. MF.PC=Peritoneal macrophages; MF.Fem.PC=female Peritoneal macrophages; MF.102+480+.PC=Large Peritoneal Macrophages; MF.AT=Adipose Tissue Macrophages; Mo.6C+II-.BI= blood Ly6C+ monocytes; Mo.6C-II-.BI= blood Ly6C- monocytes. **B)** single-cell gene expression data from bone marrow cells from the Tabula Muris (Nature, 2019) with identification of cell types (top; author-provided annotations) and expression of Pf4 projected on the tSNE plot (bottom).

2) If the authors propose a pivotal role of JAK2V617F mutation in vascular tissue-resident macrophages in the formation of aortic aneurysm, such aortopathy might be irrelevant to clonal hematopoiesis, since it has been reported that the cell origin of vascular tissue-resident macrophages is not from adult bone marrow hematopoiesis (Nat Immunol. 2016 Feb;17(2):159-68). It should be very careful to say “JAK2V617F-dependent clonal hematopoiesis drives vascular resident macrophages toward a pathogenic phenotype and promotes dissecting aortic aneurysm” in title, unless the authors provide evidence that these vascular tissue-resident macrophages are the progenies of clonal hematopoiesis.

Response: We would like to thank the reviewer for this very important point, which is fully in line with our findings. This point is clearly stated in the Introduction of the manuscript: vascular “tissue-resident macrophages, which originate from the yolk sac and fetal liver during development (Ensan et al. Nat Immunol 2016; Swirski et al. trends Immunol 2016)”. Given that Pf4 is expressed by vascular resident macrophages, but not by monocytes or bone marrow myeloid progenitors (See supplementary Fig 11 & figure 1 for reviewers), we believe that our MacR model (Pf4Cre JAK2V617Flox/lox) confirmed that Jak2 mutation predominately modulates resident macrophages phenotype towards a pathogenic profile and induces severe aortopathy.

To avoid any misunderstanding with clonal hematopoiesis of indeterminate potential (CHIP) that is also associated with the JAK2V617F mutation, reference to “clonal hematopoiesis” has been removed, keeping “JAK2V617 mutation” in the title and in the manuscript.

3) As the authors mentioned in line 415, a recent study showed that EPO injection promoted aortic aneurysm in Apoe^{-/-} mice and suggested that endothelial cell activation might be involved in EPO-induced abdominal aortic aneurysm. Whether endothelial cell activation also exists in the aortic aneurysm induced by high dose of EPO injection in the present study? If the authors could not determine the specificity of the responding cell types, I suggest this part of results to be removed from the current MS.

Response: We would like to thank the reviewer for this valid point. Indeed, we did not show the specificity of the cell types responding to EPO. However, the EPO data should be analyzed in keeping with all other experiments carried out in our study.

Data shown in figure 2 showed that genetically-induced JAK2 (mimicking EPO treatment) specifically in endothelial cells had no impact on the development of both aorta dilation and rupture. Data in figure 5 with MacR model ruled out a role for polycythemia in JAK2-related aortopathy. Given that resident macrophages specifically express EPO receptor, we speculated that high doses of EPO activated this population. We showed that EPO injection increased vascular resident cell content and local proliferation, increased local pro-inflammatory signature and induced severe aortopathy. EPO experiment provides an additional pharmacological evidence for the pathogenic effect of Jak2 engagement in these cells. To address the review’s concern, EPO data have been moved as supplemental figure 12 and a limitation has been added in the Discussion section

4) In Figure 5G, representative immunostaining of Lyve1 should be displayed.

Response: Representative immunostaining have been inserted (Fig 5J)

Reviewer #4 (Remarks to the Author):

This study tested a role for Jak2 V617F mutant form in residential macrophages in mouse aortic (ascending, descending, and abdominal) aneurysms. The role of this mutant form of

Jak2 has been tested in few other cardiovascular disease models. Overall, many results are descriptive.

Response: We would like to thank the Reviewer for his/her comments that helped improve the manuscript. Here, we used different in vivo approaches (2BMT models, 4 Cre/Lox) to decipher which cell type was involved in JAK2V617F-induced severe aortopathy. In addition, with complementary pharmacological strategies, we showed that the depletion of resident macrophages or JAK2 pathway blockade strongly attenuated local inflammatory responses and ultimately aortic disease severity. Altogether and with all due respect, we believe that our work cannot be described as mainly descriptive, but in fact provided novel important mechanistic insights into how JAK2 mutation promotes severe aortopathy and impacts on the phenotype of resident vascular macrophages.

Here are few of my concerns.

Page 6: It is unexplained why authors generated the Jak2-V617F-VE-Cadherin-Cre+/- mice. Don't we think VE-Cadherin is an EC protein? In other word, authors initially might want to study Jak2-V617F function in EC? Please provide the background information of this experiment.

Response: We fully agree with the Reviewer that VE-cadherin is almost exclusively expressed by endothelial cells after birth. However, during embryogenic stages, VE-cadherin is also expressed by non-vascular progenitors (see reference Oberlin E et al. Int J Dev Biol. 2010;54(6-7):1165–1173). To address the reviewer's concern, we analyzed mutation expression in bone marrow progenitors in JAK2V617F HC-EC model. The mutation was detected in almost all stem cells and hematopoietic progenitors. These data have been added as new supplementary fig. 2A

Figure 2: In this Figure, authors performed the BMT using bone marrows from WT mice to irradiated WT and Jak2-V617F-HC-EC mice and claimed no role of non-myeloid cells in aortic aneurysm development. Since data are all negative, it is not necessary to show these data in the main text. Authors should move the whole figure into the supplementary file.

Response: As suggested, data about the role of Jak2 mutation in non-myeloid cells were moved as supplemental fig. 6.

Data in Figure 4 did not give much useful information either, besides authors used the CITE-seq technique. It is common sense that residential macrophages present in the aortas. These data can also be in the supplementary file. Pf4 is a marker of residential macrophages. This is known. Straight use of Pf4-Cre mice will make much more sense.

Response: The aim of Figure 4 is not to show that resident macrophages are present in the aorta, but rather to compare the profile of aortic macrophages (resident or recruited) in control mice vs JAK2-V617F MyC mice with aortic dilation or aortic aneurysm. This figure provides key information, as it shows that (i) JAK2-V617F MyC aortas are enriched in a population of tissue resident macrophages with a pro-inflammatory profile and (ii) JAK2-V617F MyC aortas presenting an aneurysm further contain a population of proinflammatory recruited CCR2+ macrophages.

We agree that Pf4 has already been shown to be a marker of tissue resident macrophages. We have cited the relevant literature (Cochain et al Circ Res 2018 & Zerneck et al. Circ Res 2020). We do not claim that this is a novel finding, but we believe that these data confirm the preferential expression of Pf4 in aortic resident macrophages, which supports the use of Pf4-Cre mice to induce recombination preferentially in aortic macrophages.

Uses of the LysM-Cre+/- mice and Pf4-Cre mice to study macrophages are valid and sufficient.

However, use EPO as an activator of vascular residential macrophages is not a clean experiment. EPO receptor is expressed not only in macrophages, but also in many other cell types (PMID: 23874302, PMID: 26349059, PMID: 21357707). The results from Figure 5L to 5O were not necessary all come from macrophage activation.

Response: We would like to thank the reviewer for this point. Indeed, we did not show the specificity of the cell types responding to EPO. However, EPO data should be analyzed in keeping with other experiments carried out in our study. Data in figure 2 showed that genetically-induced JAK2 activation (mimicking EPO treatment) specifically in endothelial cells had no impact on the development of both aorta dilation and rupture. Data in figure 5 with MacR model ruled out a role for polycythemia in JAK2-related aortopathy. Given that resident vascular macrophages specifically express EPO receptor, we speculated that high doses of EPO activated this population. We showed that EPO injection increased vascular resident cell content and local proliferation, increased local pro-inflammatory signature and induced severe aortopathy. EPO experiment provide another evidence that Jak2 overactivation in tissue macrophages is pathogenic. To address the reviewer's concern, EPO data have been moved as supplemental figure 12 and a limitation has been added in the manuscript.

Results from Ki20227 in Figure 6B to 6I are also questionable, because Ki20227 not only affects residential macrophages, but also many others mechanisms (PMID: 19398755, PMID: 18085662, PMID: 22186992).

Response: We would like to thank the reviewer for this point. Indeed, Ki20227, as most of chemical agents, are not fully specific for resident macrophages. We used a protocol previously reported by Angeli's group (Lim et al. Immunity 2018) and we confirmed that Ki20227 fully depleted resident macrophages. We used alternative treatment to limit off-target effects.

As noticed by the reviewer, some studies have reported that Ki20227 may modulate angiogenesis and lymphangiogenesis. However, we do not have any evidence that impaired pathways related to angiogenesis and lymphangiogenesis are involved in the pathophysiology of dissecting aneurysm in JAK2 mutant animals. We did not find any different in HiFa, Vegf-a, Vegfr2, Vegf-C mRNA levels in the aorta between JakWT control and Jak2Myc mice. In addition, using immunostaining, we did not detect differences in VE-Cadherin expression between groups. These new data have been added as supplemental figure 10. Potential off-target of Ki-20227 has also been added as a limitation in the Discussion section.

Minor comments:

Figure 1F: why the aneurysm development differed between ascending aortas and descending aortas? Any explanation?

Response: We do not have any clear explanation for such observation. Except for HC-EC model, we found that JAKV617F induced aorta dilation in the ascending, descending aorta, as well as on the abdominal region. Aorta diameter was measured at very early stages and delayed dilatation could not be ruled out.

It is strange that the authors quantified elastin contents by the numbers of elastin layers. It makes more sense by quantify their contents by immunostaining, or count the numbers of elastin breaks.

Response: We would like to thank the reviewer for this question. The quantification of aortic wall remodeling and more specifically elastin layer degradation remains challenging and different methods are currently used by different groups. Mean layer number is quantified in routine blindly in our team by using a standardized protocol. Our strategy has been validated in several previously published works (Wang JCI 2010, Ait-Oufella ATVB 2013, Giraud Cardiovasc Res 2017, Vandestienne JCI 2021).

REVIEWERS' COMMENTS

Reviewer #1 (Remarks to the Author):

In the revised manuscript, the authors have changed their narrative from JAK2V617F-dependent clonal hematopoiesis to JAK2V617F-positive MPN in response to the comments on experimental approaches to establish a connection between JAK2V617F-dependent clonal hematopoiesis and aortic aneurysm.

Although the current study by Dr. Ait-Oufella and colleagues provide sufficient evidence to conclude that aortic aneurysm is a complication associated with JAK2V617F-positive MPN, such findings were previously reported by Yokokawa et al in *Haematologica* 2021 (PMID: 33567809; pre-published in February 2021), long before this manuscript was first submitted to *Nature Communications*.

Yokokawa et al. (PMID: 33567809) reported an association of aortic aneurysms in patients with JAK2V617F-positive MPN. They showed that hematopoietic expression of JAK2V617F promotes aortic aneurysms in mice. They also indicated an involvement of macrophages and inflammatory cytokines/signaling in JAK2V617F-induced aortic aneurysms. In addition, they showed that treatment with JAK inhibitor Ruxolitinib prevents the development of aortic aneurysms induced by JAK2V617F mutation.

The current study by Dr. Ait-Oufella and colleagues mainly validated the above findings. The new addition in this manuscript is the single-cell RNA-seq/CITE-seq analysis.

The authors should change the following terms in the manuscript since they are confusing.

Jak2V617F MyC mice. In this case, Jak2V617F flex mice were crossed with LysM-Cre mice. Myc is an oncogene that can be induced by Jak2 or Jak2V617F. Jak2V617F Myc could be viewed as Jak2V617F mice overexpressing Myc or crossed with Myc transgenic mice. In order to avoid such a confusion, the authors should use the term Jak2V617F- myeloid or Myeloid-specific Jak2V617F-expressing mice.

Jak2V617F MacR mice. In this case, Jak2V617F flex mice were crossed with Pf4-Cre mice. The authors used the term Jak2V617F MacR because they think that Pf4-Cre induces selective expression Jak2V617F in resident macrophages. This is incorrect and misleading. The authors should not argue that Pf4-Cre is selective for tissue-resident macrophages. The Pf4-Cre mouse was generated to selectively delete or express genes in megakaryocytes and platelets (Tiedt et al., *Blood* 2007; PMID: 17032923). Subsequent studies found that Pf4-Cre is not megakaryocyte lineage-specific but can be expressed in hematopoietic stem cells (HSC) and their progeny including other myeloid and lymphoid cells at significant levels (Calaminus et al. *PLOS One* 2012; PMID: 23300543). Please read these literatures carefully.

Authors wrote in page 13 (lines 299 and 300), "Previous scRNA-seq studies have identified Platelet factor 4 (Pf4) as a selective marker of aortic resident macrophages (ref. 19, 23)". These studies cited by the authors did not perform any experiment to show that Pf4 is a selective marker of aortic resident macrophages. In fact, reference #19 did not have any mention of Pf4. Reference #23 involves meta-analysis, not actual experimental analysis. However, reference #23 mentioned "Pf4, which has been considered specific for platelets and megakaryocytes, is also prominently expressed in the main population of resident vascular macrophages". Nowhere it is stated that Pf4 is a selective marker of aortic resident macrophages. The authors should not make false statement to justify the use of Pf4-Cre. They should clearly state what was found in the previous studies.

Some of the references cited in the manuscript are incorrect. The authors utilized the Jak2V617F-Flex mouse in this study. The reference #43 is not the correct reference. The original paper that reported the Jak2V617F-Flex mouse is Salma Hasan et al. *Blood* 2013 (PMID: 23863895).

The reference #44 cited for Pf4-Cre mouse is also incorrect. The correct reference for Pf4-Cre mouse is Tiedt et al., Blood 2007 (PMID: 17032923).

Reviewer #2 (Remarks to the Author):

Al-Rifai et al. revised their manuscript adequately, and demonstrated perivascular tissue-resident macrophage with Jak2VF mutation promoted deleterious aortic wall remodeling and dissecting aneurysm. I have no more queries.

Reviewer #3 (Remarks to the Author):

The authors have addressed all my concerns. I have no more questions.

Reviewer #4 (Remarks to the Author):

Authors addressed all my prior concerns.

Responses to Reviewer #1

In the revised manuscript, the authors have changed their narrative from JAK2V617F-dependent clonal hematopoiesis to JAK2V617F-positive MPN in response to the comments on experimental approaches to establish a connection between JAK2V617F-dependent clonal hematopoiesis and aortic aneurysm. Although the current study by Dr. Ait-Oufella and colleagues provide sufficient evidence to conclude that aortic aneurysm is a complication associated with JAK2V617F-positive MPN, such findings were previously reported by Yokokawa et al in *Haematologica* 2021 (PMID: 33567809; pre-published in February 2021), long before this manuscript was first submitted to Nature Communications.

The current study by Dr. Ait-Oufella and colleagues mainly validated the above findings. The new addition in this manuscript is the single-cell RNA-seq/CITE-seq analysis.

Response: We would like to thank the reviewer for his/her comments which helped improve the manuscript. However, with all due respect, we disagree with the reviewer's comment concerning the lack of novelty of our work. In the study by Yokokawa (Haematologica. 2021 Jul 1;106(7):1910-1922) the authors used a transplantation model of JAK2V617F bone marrow cells in hypercholesterolemic Apoe KO mice infused with angiotensin II. As the reviewer certainly knows, this model is very far from the clinical setting and is non-clinically relevant. Moreover, and more importantly, this study did not provide any mechanistic insights into how JAK2V617F mutation can promote aortopathy, and in particular made no mention of the central role of resident macrophages in JAK2V617F-induced severe aortopathy.

Please note that in our study mice expressing the human JAK2V617F mutation spontaneously developed dissecting aortic aneurysm in the absence of AngII challenge. In addition, we identified for the first time the critical impact of JAK2V617F mutation on vascular resident macrophages functions. Such results were obtained using different in vivo approaches, including three Bone Marrow Transplantation models and four 4 Cre/Lox models. Finally, with complementary pharmacological strategies, we showed that the depletion of resident macrophages or JAK2 pathway blockade strongly attenuated local inflammatory responses and ultimately aortic disease severity.

The novelty of our work has been highlighted in the discussion section

The authors should change the following terms in the manuscript since they are confusing. Jak2V617F MyC mice. In this case, Jak2V617F flex mice were crossed with LysM-Cre mice. Myc is an oncogene that can be induced by Jak2 or Jak2V617F. Jak2V617F Myc could be viewed as Jak2V617F mice overexpressing Myc or crossed with Myc transgenic mice. In order to avoid such a confusion, the authors should use the term Jak2V617F- myeloid or Myeloid-specific Jak2V617F-expressing mice.

Response: As suggested, Jak2V617F Myc has been replaced Jak2V617F Myel.

Jak2V617F MacR mice. In this case, Jak2V617F flex mice were crossed with Pf4-Cre mice. The authors used the term Jak2V617F MacR because they think that Pf4-Cre induces selective expression Jak2V617F in resident macrophages. This is incorrect and misleading. The authors should not argue that Pf4-Cre is selective for tissue-resident macrophages. The Pf4-Cre mouse was generated to selectively delete or express genes in megakaryocytes and platelets (Tiedt et al., *Blood* 2007; PMID: 17032923). Subsequent studies found that Pf4-Cre is not megakaryocyte lineage-specific but can be expressed in hematopoietic stem cells (HSC) and their progeny including other myeloid and lymphoid cells at significant levels (Calaminus et al. *PLOS One* 2012; PMID: 23300543). Please read these literatures carefully.

Response: We agree that expression of Pf4 is not entirely restricted to bona fide tissue resident macrophages but is also found (albeit at a lower level) in other tissue infiltrating macrophages. We have reworded the text to indicate that this model represents rather a strategy to induce preferential recombination in tissue macrophages. However, the main reason for using this approach is that Pf4 is not expressed in circulating monocytes or their progenitors. Notably, we did not detect Pf4 in circulating monocytes by scRNA-seq, as shown in supplementary Figure 11. Interrogation of bulk RNA-seq data from ImmGen also indicated that Pf4 is not expressed in circulating monocyte,s but is highly expressed in tissue resident macrophage subsets from the peritoneum and adipose tissue. Pf4 is also not detected

in monocyte progenitors in the bone marrow. The preferential recombination pattern in tissue resident macrophages vs. circulating monocytes in Pf4-cre mice has been characterized in the literature (see McKinsey, 2020; Abram, 2014). Therefore, together with our other strategies to induce the JAK2V617F mutation in myeloid cells (Myel Model), the Pf4-cre model provides strong additional evidence that the observed aortopathy phenotype was promoted by tissue macrophages and not by classical monocytes and their progenitors.

To address the reviewer's concern and to avoid confusion, Jak2V617F MacR mice has been replaced by Jak2V617F Pf4 mice.

Authors wrote in page 13 (lines 299 and 300), "Previous scRNA-seq studies have identified Platelet factor 4 (Pf4) as a selective marker of aortic resident macrophages (ref. 19, 23)". These studies cited by the authors did not perform any experiment to show that Pf4 is a selective marker of aortic resident macrophages. In fact, reference #19 did not have any mention of Pf4. Reference #23 involves meta-analysis, not actual experimental analysis. However, reference #23 mentioned "Pf4, which has been considered specific for platelets and megakaryocytes, is also prominently expressed in the main population of resident vascular macrophages". Nowhere it is stated that Pf4 is a selective marker of aortic resident macrophages. The authors should not make false statement to justify the use of Pf4-Cre. They should clearly state what was found in the previous studies.

Response: In ref 19 (Cochain et al. Circ Res 2018), Pf4 is highly expressed in resident macrophages (please double check figure 2 of their work). The reference 23 has been replaced by a new one (McKinsey, elife 2020).

Some of the references cited in the manuscript are incorrect. The authors utilized the Jak2V617F-Flex mouse in this study. The reference #43 is not the correct reference. The original paper that reported the Jak2V617F-Flex mouse is Salma Hasan et al. Blood 2013 (PMID: 23863895).

Response: The modification has been done accordingly.

The reference #44 cited for Pf4-Cre mouse is also incorrect. The correct reference for Pf4-Cre mouse is Tiedt et al., Blood 2007 (PMID: 17032923).

Response: The modification has been done accordingly.